# Jupyter Book as an open online teaching environment in the geosciences: Lessons learned from Geo-SfM and Geo-UAV

Peter Betlem[1, 2, 3], Nil Rodes[1], Sara Mollie Cohen[1], and Marie A. Vander Kloet[1, 4]

[1]The University Centre in Svalbard, Longyearbyen, Svalbard, Norway
[2]Norwegian Geotechnical Institute, Oslo, Norway
[3]Department of Geosciences, University of Oslo, Oslo, Norway
[4]Department of Education, University of Bergen, Bergen, Norway

**Correspondence:** Peter Betlem (peter.betlem@ngi.no)

**Abstract.** Together with our students, we co-created two open-access geoscientific course modules using the Jupyter Book environment and appraised the environment's suitability for co-creation and open learning. The modules were iteratively revised over a four-year period and covered the acquisition of drone data and subsequent processing of digital outcrop models. Each module implemented an in-line collection of videos, animations, code snippets, slides, and interactive material to complement the main text in a diverse open learning environment. Behind the scenes, was used to facilitate content versioning, co-creation and open publishing of the resources. We found that students approved the framework and especially valued the framework's accessibility, inclusivity, co-creation capabilities, interactivity, and use of animations and multimedia. Collaboration certainly helped cultivate an interest in both student and instructor to revise the source materials and updating information where it was deemed outdated or unclear, regardless of the contributor's background, affiliation or level of experience. However, we found that effective co-creation relied on students being familiar with the tools at their disposal, plus be given the opportunity to contribute in their own ways. Through our combined efforts, we succeeded in providing lasting, up-to-date and open course materials to a campus with a small department that does not have significant experience nor capacity in developing and maintaining open educational resources. Finally, although we established their use, work remains to establish optimal implementations for educational GIFs.

## 1  Introduction

The concept of openness and sharing is central to many disciplines, particularly in teaching, research, and software (Khan and Ur Rehman, 2012; Hockings et al., 2012; Abernathy, 2023; Jhangiani and Biswas-Diener, 2017). Within education specifically, open pedagogy (OP) envisions a more democratic, accessible, and affordable learning environment by using open educational resources (OERs) and avoiding expensive proprietary materials (Wiley and Hilton, 2018; Abernathy, 2023; Wiley and Hilton, 2018; Christiansen and McNally, 2022; Harrison et al., 2022; Matkin, 2009). In so doing, OP emphasizes transparency, collaboration, and student-driven learning (Hegarty, 2015).

Despite its growing adoption, OP remains far from a formalised standard. Rather, it is a set of aspirational guidelines encouraging the creation, adaptation and sharing of educational materials (Wiley and Hilton, 2018; Christiansen and McNally,

2022; Tietjen and Asino, 2021; Weller, 2014). Herein OERs and OER-enabled pedagogy (OER-P) play an important role and have seen an update in recent years as an alternative to conventional scholarly and educational publishing (Tietjen and Asino, 2021; Wiley, 2013; Wiley and Hilton, 2018). Through retention, reuse, revisition, remix, and redistribution, OERs have the unique opportunity to deliver inherently collaborative, transparent workspaces and innovations that extend beyond the original authoring institution or idea (Audrey Azoulay, 2019; Caswell et al., 2008). OERs also inherently enhance the visibility and accessibility of educational content, promoting wider participation (Jhangiani and Biswas-Diener, 2017; Barba et al., 2019).

Certainly, OERs are hardly new to education; however, what could or should "count" as OERs has become a source of concern for scholars and advocates who note the casual use of the term "open" for materials that neglect or obstruct the 5Rs of OER (typically because of copyright restrictions) (Wiley and Hilton, 2018). It can be useful then to consider how "openness' can be understood and assessed, which should ideally be in tandem by both educators and students.

Importantly, co-creating OERs with students increases diversity in teaching materials, enhances engagement and improves learning outcomes (Biddle and Clinton-Lisell, 2023; Lambert, 2018; Kelly et al., 2022; Nusbaum, 2020). Overlapping with scholarship on Students as Partners, OER-P strategies enable what Bovill and Woolmer (2019) describe as co-creation *in* curriculum and co-creation *of* curriculum. This approach balances power dynamics between teachers and students, reframes knowledge and knowledge production, and "counters the increasing commodification of learning" (Bovill and Woolmer, 2019, p. 408). It is from this point, that our project emerges - we are curious about the potential of OER resource development as a transformative pedagogical practice that is undertaken collaboratively with students.

## 1.1 Jupyter Books as a tool for OER development

Today, OP and OER-P benefit from a rich ecosystem of open tools like Project Jupyter, which promotes open standards and checks many of the openness requirements (Project Jupyter, 2023; Granger and Perez, 2021). Jupyter helps decompose problems and tell stories with code and data through a range of tools of which the computational Jupyter Notebooks are perhaps most famous (Granger and Perez, 2021; Project Jupyter, 2023). It is thus not surprising that it is used widely in data science, machine learning, scientific computing, and even teaching. Recently, the Jupyter Book environment has emerged as an extention that extends the computational Notebook environment with narrative and multimedia content (Executable Books Community, 2020). Simply put, Jupyter Book provides an interface for building publication-ready books ("Jupyter Books") that seamlessly integrate computational (e.g., Jupyter Notebooks, programming scripts) and narrative (e.g., text files, images, videos) content (Executable Books Community, 2020). The resulting user-editable "unbooks" (Woodworth, 2011) integrate easily with co-creation and version control solutions like git and are ideal for open publishing.

In this contribution, we test whether Jupyter Books can indeed act as a diverse, equitable, and inclusive learning environment, embracing the three pillars of "open" social justice: redistributive, recognitive, and representational (Lambert, 2018; Biddle and Clinton-Lisell, 2023). First, we evaluate the pedagogical potential of co-created Jupyter Books, evaluate the resource's openness, and assess our students' learning experiences. Second, we assess whether the Jupyter Book/ framework and co-creation can be used for OER development with only limited resources. Third, we appraise student reception to the multimedia environment, as well as optimal playback times versus student retention to optimise the use of animation versus videos in

future module designs. In order to accomplish the above, this manuscript first documents the implementation of Jupyter Book and in the design of two geoscience undergraduate modules on unmanned aerial vehicle (UAV) data acquisition and structure-from-motion (SfM) photogrammetry processing, respectively, as part of a transition to OER-P teaching at a small campus. It then demonstrates the openness and accessibility of the framework (including the use of animations), assesses user and student learning experiences, and appraises the framework's co-creation possibilities.

## 2 Methods and data

### 2.1 Context and participants

This study was conducted over 4 years as part of two geology courses at the University Centre in Svalbard, a small public university centre in northern Norway. Both courses were taught asynchronously during a one-week interval by the same instructors, with all materials provided online. Class sizes ranged from 10 to 20 participants with diverse Earth Science backgrounds.

Course 1: An annual undergraduate geology course focusing on geoscientific digital techniques (n=62 over four years). Activities included digital field notebooks, data acquisition, geological model generation, and multi-physical data integration. Participants were primarily western European and Scandinavian students, requiring at least 60 ECTS in natural science, including 30 ECTS in geosciences.

Course 2: A multidisciplinary short course (n=10) on UAV-based data acquisition and processing, offered in summer 2023. Participants had diverse educational backgrounds, including scientific and technical staff, and students from various science, technology, engineering, and math (STEM) fields.

The Geo-SfM module (Betlem and Rodes, 2024) was implemented as part of Course 1 in 2021, initially taught digitally due to COVID-19 and redesigned from a teacher-centric module taught previously. It introduces structure-from-motion photogrammetry and provides detailed best practices. Subsequent years saw in-person teaching with minor revisions based on colloquial and questionnaire feedback.

Course 2's syllabus includes the Geo-SfM and Geo-UAV modules. The Geo-UAV module (Rodes et al., 2024) teaching UAV-based data acquisition and processing, providing self-explanatory recipes and tutorials on legal frameworks, piloting, and data acquisition. Both modules were developed from experiences and best practices from the Svalbox project (Senger et al., 2021; Betlem et al., 2023). Fieldwork tested the portability of Geo-UAV, implementing either the online tutorials or exported PDFs while teaching in the field.

### 2.2 Module and course design

We designed the Geo-SfM and Geo-UAV modules to facilitate an inclusive, accessible, and diverse learning environment. Our design drew inspiration from textbooks and tutorials using Sphinx and Jupyter Book (Henrikki Tenkanen et al., 2023, 2022; Lehmann, 2011; Executable Books Community, 2020; Rhoads and Gan, 2022; Community, 2022), which integrate interactive

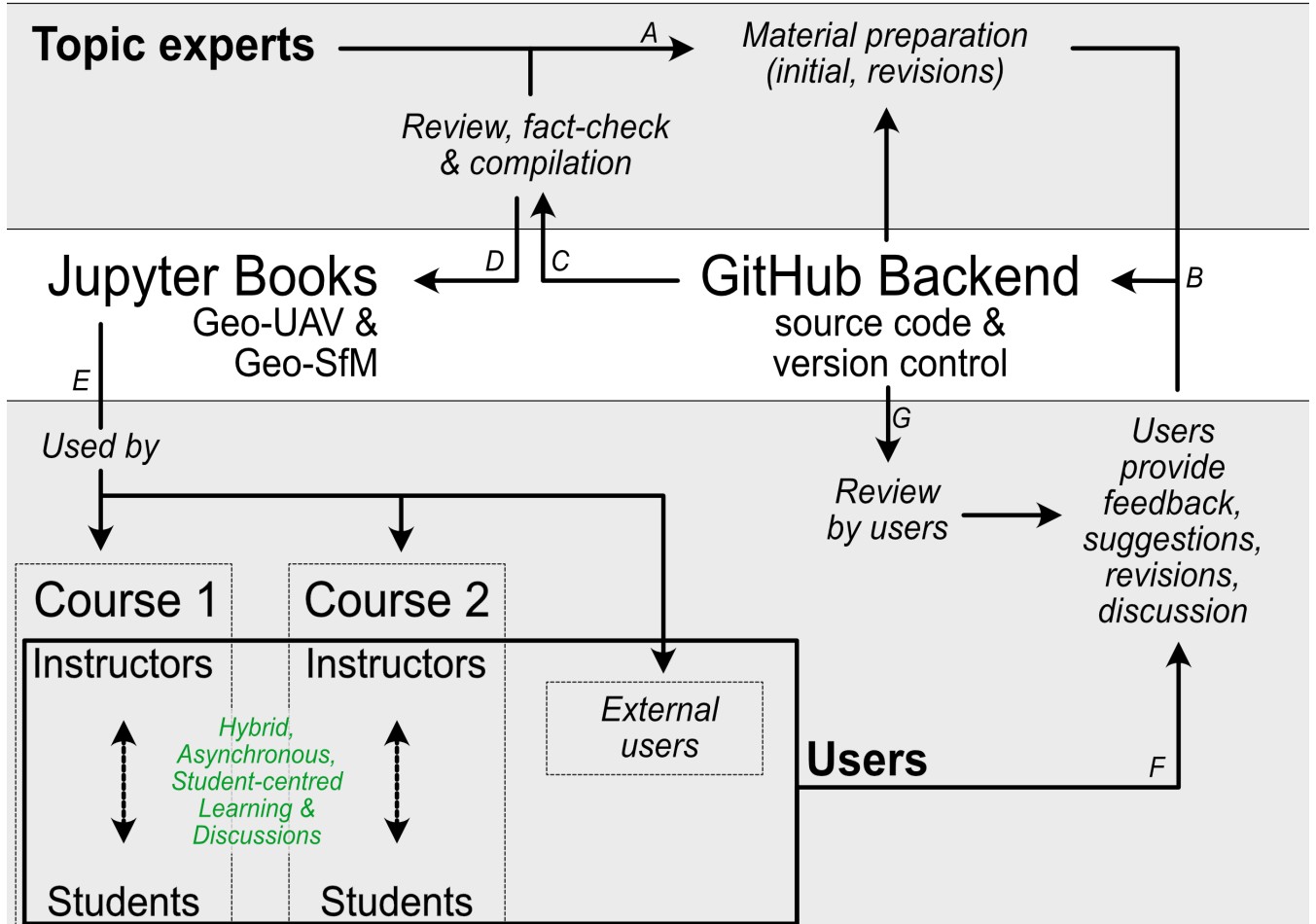

**Figure 1.** Instructional approaches of Geo-SfM and Geo-UAV integrating the backend for co-creation. Topic experts prepared the initial material (A), made it accessible through (B), and compiled the Geo-UAV and Geo-SfM Jupyter Books (C, D). These were subsequently used in Courses 1 and 2, as well as by external users (E), all of whom were invited to provide feedback, suggestions, and to implement revisions (F). Review of the latter was done by both expert (C→A→B) and user groups (G→B), with re-compilation (D) done after final review (C) by a topic expert, before repeating as necessary.

components and narrative content. Jupyter Book was chosen to integrate all course content, with sessions increasing in difficulty
and depth, including introductions, background information, multimedia content, tutorials, and assignments. Mini-lessons on project management, data structuring, and automation were also included.

Following an introduction to the layout of the modules, sessions and key learning outcomes, students were introduced to the platform (i.e., the backend) and requested to sign up and raise a simple welcome/"hello world" issue through one of the on-page menu bars at the start of the respective courses. This was done to facilitate optimal use of the collaborative framework
and allow the students to familiarise themselves with the backend, including issue tracker and online feedback solutions. The

students were then asked to work through the course modules in pairs, applying the concepts of pair learning to further enhance collaborative learning (Nagappan et al., 2003; Drey et al., 2022).

The platform, including its Classroom tools, has previously been shown to improve the educational experience for students and teachers (e.g., Zagalsky et al., 2015; Fiksel et al., 2019), and facilitates open hosting of documentation. The use of allowed detailed tracking of suggestions and corrections proposed by the students and other participants, thus forming the backbone to the co-creation and cooperative learning framework. This detailed log of "improvable" sections (e.g., changes in course content, more accessible phrasing, and additional/revised visual and multimedia assets) was used to further diversify the teaching material and adapt content to the styles and needs of the students. As instructors, we held few in-person lectures and were mainly present to facilitate discussions, guide asynchronous learning and provide technical support (Fig. 1).

Starting in 2024, we offered a more extensive, preparatory three-hour tutorial on contributing through forks and pull-requests following feedback from the 2021 and 2022 courses. These tools allow sophisticated changes to the source code and greatly expand upon how contributions can be made, yet require an extended introduction for optimal use. Each pull-request interaction is documented, attributing co-creators to the revised resource as a form of ownership. We asked students to review each other's proposed revisions and additions prior to final approval by instructors and experts (Fig. 1).

The platform provided an alternative venue to ask questions and students were encouraged to seek and receive feedback through the platform as well as from instructors. Online (issue) participation on , discussions and physical presentations replaced graded assessments and exams. Classroom teaching further implemented the colloquial sharing of results and experiences during daily recaps in which students presented both their results and stumbles, with feedback and possible solutions mostly provided by other working groups. Peer-to-peer evaluation was also encouraged for pull-requests and revisions suggested to the courses, though were not part of the grading process. The setup of the modules, implementing gradual and asynchronous learning, naturally facilitated grading through module completion and participation. In Course 2, the shared assessment for the individual sessions was certified and documented in a course certificate, listing the accomplished learning objectives, and stating their equivalent.

GIFs, given their capacity to capture short animations and generally small file sizes, have become a key communication tool on par with other visual media (Bakhshi et al., 2016; Miltner and Highfield, 2017) and their inclusion has been shown to increase engagement and lower the barriers for participation (Bakhshi et al., 2016). For this reason, we implemented both shorter and longer animations to supplement videos, detailed plain-language summaries and static figures in order to improve the accessibility of learning materials. We used the LICEcap library (Frankel, 2023) for simple animated screen captures because it is a lightweight, intuitive, and flexible application that supports both Windows and OSX operating systems. The library supports custom capture windows, intermittent recording, and on-screen text messages and information. In total, 31 looping animations were incorporated with durations of between 3.8 and 78 seconds (Table S1). Videos were mainly recorded through the Open Broadcaster Studio (OBS) software package (Kristandl, 2021; Bailey et al., 2017). OBS Studio is a free and open-source software that is a reliable tool for the recording of screens, (instructional) videos and online streams and is easily used without formal training (Basilaia et al., 2020). OBS Studio supports screen, window and camera recording with configurable audio input and output. In total, 11 videos were incorporated with runtime durations of between 39 seconds and

6:28 minutes (Table S1). Students were also shown how to use the software, to lower the barrier for co-creation of multi-media assets.

During the development stages, we particularly appreciated the rich documentation provided by the Jupyter Book project pages (Executable Books Community, 2020) that offer a detailed tutorial of what is possible with the Jupyter Book framework and provide an extensive step-by-step guide on how to get started. This easy-to-follow guide further details various options for sharing the dynamic pages, which are optimised for both mobile and desktop use, and even allowed module participants to make more sophisticated changes to the modules. The runtime environment needed to compile the modules can be easily installed using the standard Python package managers pip, conda or mamba, and contains a set of command-line utilities for the compilation of textbooks from Markdown text (.md), Jupyter Notebook (.ipynb) or reStructuredText (.rst) files – all of which open formats. The implementation of the Markedly Structured Text (MyST) syntax, an extension of Markdown, provides simplicity while still being powerful enough to create rich content pages with text, figures, automatically-generated citations, executable and in-line code-cells, slide-shows, and embedded files (e.g., three-dimensional [3-D], interactive environments) and videos (Chen and Asta, 2022; Executable Books Community, 2020). Although not explored in the Geo-SfM and Geo-UAV modules, pages can also integrate with cloud-providers such as JupyterHub (Project Jupyter, 2023) and Google Colab (Bisong, 2019) to facilitate executable and programmable content without having to install libraries locally.

## 2.3  Open Pedagogy study

The conducted pedagogy study can be divided into two phases: the initial design phase and the testing phase. During the design phase of the Geo-SfM module in 2021 and 2022, we collected qualitative data from course evaluations and in-class feedback sessions (n=32). Students' feedback optimized the Geo-SfM module for the following years and informed the design of the Geo-UAV module in early 2023. Starting in 2023, we also implemented a student questionnaire to gather quantitative and qualitative data on students' experiences and the modules' perceived impact on their learning.

The questionnaire (Table S3) focused on the user and learning experience, platform accessibility, multimedia and content diversity, and options for student co-creation. First, students provided information on their educational backgrounds, and prior-knowledge self-assessments on their programming experience, use of Project Jupyter tools, online documentation, video hosting platforms, and animated GIFs. Second, they answered quantitative (5-point Likert scale; Fig. 3) and qualitative (Fig. 2) questions about the integrated Jupyter Book and platforms, as well as the integration of multimedia such as GIFs and videos. The latter specifically addressed different playback durations of the implemented animations and videos to assess student reception and determine optimal playback times versus student retention. Qualitative feedback was categorized as either *constructive criticism* or *positive feedback*.

The questionnaire was developed with the Norwegian National Ethics Committee's Guidelines for Research Ethics in the Social Sciences and Humanities (NESH, 2024) in mind. Further, the study was internally reviewed through the University Pedagogy Programme at the University Centre in Svalbard, i.e., the study's host institution. Participation in the survey was voluntary, anonymous, and without rewards. The survey was made available through the Jupyter Book modules, and students completed the survey online via Nettskjema. Nettskjema is an online survey tool developed by the University of Oslo and

165 is specifically designed to meet Norwegian privacy requirements (Engh and Speyer, 2022). We also collected feedback from external participants who accessed the online modules independently throughout 2023.

## 2.4 Evaluating openness

To grade the openness and accessibility of the two course modules and the Jupyter Book/ framework as a whole, we implemented the *Open Enough* rubric proposed by Christiansen and McNally (2022). McNally and Christiansen (2019) suggest that the

170 openness of OERs can be evaluated through the eight primary factors of openness, being copyright, accessibility, language, support costs, assessment, digital distribution, file format, and cultural considerations, each ranked from "most open" to "closed". However, we evaluated *Harvestability* as a *Technical* rather than *Pedagogical* factor, and based the ranking on a combination of colloquial and questionnaire student feedback, as well as on our own observations as educators and instructors.

## 3 Results

In 2023 and 2024, students participated in the questionnaire during dedicated timeslots immediately after the Geo-SfM module in Course 1 (n=30) and at the end of Course 2 (n=10). Out of 40 students surveyed, 36 responded. Additionally, four external participants independently responded, resulting in a total of 40 responses. We created the initial coding scheme for qualitative feedback by screening all responses for common themes and understanding levels (Taylor et al., 2015). Table 1 lists the coding scheme and student responses for each category.

**Table 1:** Qualitative student feedback with descriptions and examples, grouped by category.

| Code categories | Code | Description | Example |
|---|---|---|---|
| Accessibility, content and language | Constructive criticism | Responses that criticised the navigation and design of the modules. N=18 | Instructions were sometimes not 100 % clear If there would be a search tool, it might be easier to find information on the page.    Other languages than English    Maybe sometimes background information and instructions are a bit mixed up.    Sometimes the background context was lacking, meaning the tutorial was very helpful itself but it required prework that was not explained.    Some tricks and tips were not in the Compendium |
| Accessibility, content and language | Positive feedback | Responses that positively referred to the accessibility, content and language of the modules. N=26 | While the tutorial explained exactly what to click it also explained why, which was helpful and gave context.    I liked how open and accessible everything was, all the supportive python codes etc., just there to use and make life easier.    I really liked how clear and step-by-step the instructions were, as it made it easier to move forward (and go back) in my own pace.    The use of alternative/multimedia learning resources makes it inclusive.    It is a very useful resource.    I will always use it when working with photogrammetry. |
| Co-creation | Constructive feedback | Responses that independently referred to aspects of co-creation. N=8 | It is good that changes can be put in very easy by the user.    I liked that it was interactive and that you could change or add anything to improve it for next year.    Also, being able to make small changes to the actual site felt inclusive.    Some of the instructions used words/names from previous versions of Agisoft, but then again we were encouraged to edit this ourselves (a good thing). |
| Technical aspects: Navigation and design | Constructive criticism | Responses that critisised the navigation and design of the modules. N=16 | Navigation is not intuitive.    The flow of the page is not great.    Links referring to other compendiums was confusing in the beginning Sometimes a bit too much text and therefore loss of structure. |

| Code categories | Code | Description | Example |
|---|---|---|---|
| Technical aspects: Navigation and design | Positive feedback | Responses that positively referred to the navigation and design of the modules. N=28 | Flows really well.    Clear and logical breakdown of processes and steps are well explained.    I liked that the processes had been broken down into bitesize chunks and the exercises were logical to follow. |
| Technical aspects: Multimedia integration | Constructive criticism | Students were specifically asked about the things they disliked about the use of multimedia in the compendiums. N=30. | In some of the videos the text was so zoomed out that it was hard to see what exactly what was being done.    Videos were too slow Sometimes not text to describe the step, only GIF.    Provide text alongside animations/videos.    Not able to pause GIFs.    GIFs do not have a clear start/end Some GIFs were a bit too long, so if you missed something in the beginning you had to re watch |
| Technical aspects: Multimedia integration | Positive feedback | Students were specifically asked about the things they liked about the use of multimedia in the compendiums. N=39. | The use of videos throughout and along with the instructions was good.    Provide quick overview.    Made things easy to follow, findable in menus.    GIFs are short and therefore show the information very effectively.    I did not watch as many YouTube videos but they can show more complex things.    As I am a visual learner, the animated GIFs helped me a lot throughout the week as it helped to navigated what needed to be done. |

The quantitative results are shown as stacked box-plot charts for either module (Fig. 2, Fig. 3), with examples of student responses from open-ended questions included in the results and discussion. The analysis does not distinguish between internal and external evaluations, nor does it separate results by course.

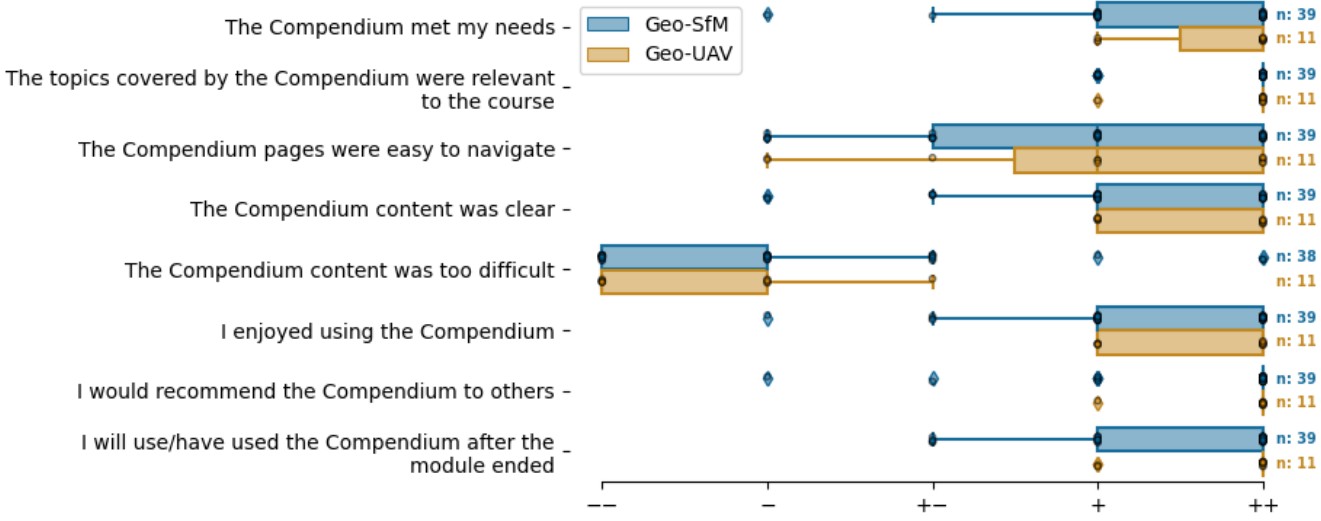

**Figure 2.** Quantitative student feedback on the Geo-SfM and Geo-UAV modules, here referred to as Compendiums.The bars, boxes and whiskers indicate the mean, one standard deviation and two standard deviations, respectively. Individual scores are separated for clarity.

## 3.1   Student perceptions on the learning environment

Student perceptions of the Geo-SfM and Geo-UAV modules were measured using Likert-scale questions developed specifically
for this study, with feedback largely similar between the two modules. Overall, students agreed that they were excited about

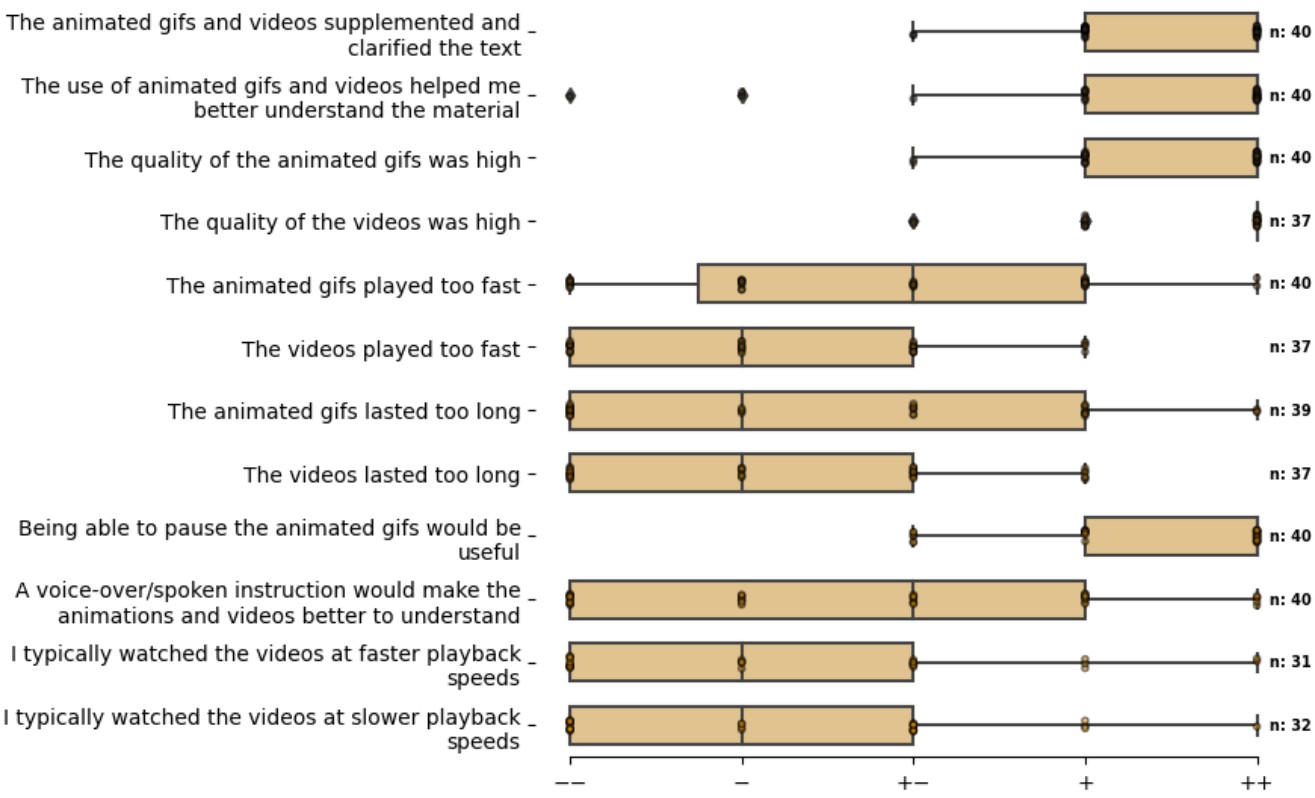

**Figure 3.** Student feedback on how well they experienced the inclusion of animations and video assets. The bars, boxes and whiskers indicate the mean, one standard deviation and two standard deviations, respectively. Individual scores are separated for clarity.

using the online modules, that the modules met their needs, and that the content was clear and easy to navigate. Students also indicated they would recommend the modules to others and use them as reference works in the future (Fig. 2).

Answers to the open-ended questions (e.g., Table 1) reflected a positive learning experience. Students valued the Jupyter Book/ implementation for its modernness and clear structure, despite few having had prior familiarity with it or other documentation platforms like it (Fig. S1). They also appreciated the platform's open online nature, which was mentioned to facilitate diverse and asynchronous learning at their own pace.

Students praised the Geo-UAV module for providing a "very good overview of a complex topics and integration of different sources" and "liked how open accessible everything was". They appreciated "that the processes had been broken down into bitesize chunks and the exercises were logical to follow". One student even referred to the modules "as a 'bible' of tutorials throughout the course", while another noted that the platform helped "consolidate a large amount of information that, if it had purely been communicated verbally, would have been overwhelming to absorb".

Similar reflections were obtained for the Geo-SfM module. Students noted that "all the supportive Python codes etc., [are] just there to use and make life easier" and "liked that pictures and GIFs were used in the tutorials", though not all students were equally excited about lengthy animations.

## 3.2 Student perceptions on integrated multimedia use

As instructors, we had aimed to create a diverse and accessible learning environment through use of multimedia integration and student-led content creation. Thus, students were specifically asked about their previous experiences with multimedia (Fig. S1) and how they perceived the use of GIFs, videos, and interactive content in the modules.

Students highlighted the benefits of animations and videos alongside text descriptions, noting these elements enhanced course content diversity and accessibility. Their open-ended remarks (Table 1) on the use of animations and videos within the modules aligned with their quantitative feedback (Fig. 3). They agreed that animations and videos supplemented the main text effectively and were of high quality. However, students found the playtime of multi-step animations (i.e., GIFs) too long and suggested a pause function (Fig. 3). Open-ended responses indicated frustration with waiting for GIF loops to end and the need to replay them multiple times to understand all steps, as shown in student-reported playtime statistics (Table S2). Examples include that they did not like having "to wait for the loop to end to see again the info [they] wanted to see" and that they "had to play it [GIFs] several times to identify all steps". Despite this, students found GIFs useful for illustrating processes and reducing reading.

## 3.3 Student perceptions on co-creation possibilities

Although student perception on co-creation was not quantitatively assessed, eight students independently reflected on it through the open-ended survey questions. Students noted that "being able to contribute to it [the modules]" and "also see other's contributions was helpful in filling in gaps".

Students actively enhanced the modules by extending functionality, improving clarity, and updating animations and figures (Fig. S2). This is evidenced by 39 pull requests to the Geo-SfM module by 10 students from the 2024 class, who benefited from an extended introduction to . Contributions varied from single-word edits to multi-paragraph revisions and new animations.

Unsurprisingly, a subset of students in prior years reported agreement that they were "a bit confused . . . when it came to using " as they were not fully introduced to the platform's possibilities at the onset of the courses. The differing levels of introduction, however, did not change student-reported inclusiveness in content creations, or their overall learning experience. Both cohorts reported that it felt inclusive to learn from student-proposed changes from previous years and to be able to further revise and improve the resources for future use, thus becoming part of the community. The "use of /git to enable community contributions" was noted as an important factor that set the modules apart from previous learning experiences.

## 3.4 Open Enough

Both Geo-UAV and Geo-SfM (and the Jupyter Book/ framework as a whole; Table 2) rank high on openness within the *Open Enough* rubric, outranking many of those rated by Christiansen and McNally (2022). Key contributions are the modules' learner-centred design and the implementation of collaborative and inclusive design choices in a modern, open format. The few *Mixed* and Closed ratings are a result of design choices, such as the use of expensive UAVs and sole implementation of the English language.

**Table 2:** Openness as evaluated against the *Open Enough* considerations outlined by Christiansen and McNally (2022), treating *Harvestability* as a *Technical* rather than *Pedagogical* factor.

| | **Technical Factors** | | | |
|---|---|---|---|---|
| Course Module | Copyright/OL | Discoverability | File Format | Harvestability |
| Geo-UAV | Mixed (CC-By-NC) | Most Open | Most Open | Most Open |
| Geo-SfM | Mixed (CC-By-NC) | Most Open | Most Open | Most Open |

| | **Pedagogical Factors** | | | |
|---|---|---|---|---|
| | Language | Material costs | Assessment | Accessibility |
| Geo-UAV | Closed | Mixed | Mixed | Most Open |
| Geo-SfM | Closed | Mixed | Mixed | Most Open |

| | **Other considerations** | | | |
|---|---|---|---|---|
| | Diverse users | Culturally inclusive | Easy to navigate | Responsive design |
| Geo-UAV | Yes | Yes | Yes | Yes |
| Geo-SfM | Yes | Yes | Yes | Yes |

## 4 Discussion

Openness and interactivity drives engagement, interest, and exploration of concepts, which is crucial to both learning and scientific thinking. Both Geo-UAV and Geo-SfM were designed with that in mind, and both were tailored bottom-up to support courses where students have a wide range of experiences and abilities. Jupyter Books are naturally suited for such an environment. The framework provides a way to integrate extensive narrative content with examples and code templates for those in need of support, while more-experienced students can modify and adapt examples to independently explore more advanced scenarios.

Simultaneously building comprehensive teaching materials and designing pedagogical feedback processes, however, can be a challenging task, and one that can only be accomplished through an interdisciplinary collaboration between scientists, social scientists, and students. Over these past four years, we learned that from the iterative development of the modules and courses, as well as from designing the pedagogics framework itself. Initially, the focus was on assessing the technical usability of the modules and assessing the usability of the Jupyter Book framework for its learning-potential, including the role of integrating multimedia and animations therein. In years 3 and 4, qualitative student reflections highlighted the potential for co-creation and the inclusivity, diversity, and accessibility benefits of the Jupyter Book/ framework. These insights lay the groundwork for

future activities to quantify student perceptions of these aspects. Our results provide a starting point and valuable insight into designing and co-creating future OER-P content using modern educational platforms. Overall, students perceived the Jupyter Book format and modules as useful for supporting their learning, while also expressing some concerns about some of the design choices. Many of these have been systematically addressed during the 4-year runtime of the project, in part through student contributions, in part through social science insights.

In the discussion that follows, we integrated the students' survey responses with our own observations to evaluate the modules' relative openness/accessibility and other pedagogical factors by implementing an *Open Enough* rubric (Christiansen and McNally, 2022). We did so to address the objectives raised in the introduction, as well as to aid our understanding of how the Jupyter Book framework is viewed by students and how it can be implemented as a means of co-creative open learning.

## 4.1 Learner-centred design - Co-creating accessible and diverse resources

Open-source curricula have been shown to facilitate participation, discussion and co-ownership amongst students and the broader community, inviting all to participate in the collaborative development of educational resources (Chen and Asta, 2022; Kim et al., 2021). Analysis of colloquial and quantitative questionnaire feedback (Table 1) provided by the students indicated as much and highlighted several advantages of using the Jupyter Book/ framework, in particular. First, the interactivity of the modules, exposure to pre-written code (snippets), and integrated multimedia use provided a rich and diverse learning experience certainly helped demystify the abstract notions of scientific data acquisition and processing. Second, we noticed that the availability of co-creation examples from previous years to learn from, as well as being introduced to the unformatted source code of the teaching resources lowered the barrier for students to become contributors. Third, students noted the learning effectiveness of the modules, in particular the usability of step-by-step instructions that were provided in various formats, different voices, and different levels of interactivity. Fourth, students affirmed what we had hypothesised – that for students to become contributors, they first need to be comfortable using the tools and be given ample opportunity and freedom to revise content, with the side note that it is reviewed and fact-checked by other students and course instructors prior to implementation. The latter, however, must not stand in the way of students to think about what else can be built into the tool to support their learning and that of others. Indeed, students agreed that the exposure to code, programming and the backend was beneficial to the learning experience (Tab. 1), which may in part be because creating cohesive content follows aspects of (scientific) problem solving: Decomposition, Pattern recognition, Abstraction, and Algorithm design (Barba et al., 2019).

Given that the modules are openly available on the internet and provide accessibility by supplementing multimedia and user-interactions, it is not surprising that the students rated the Geo-UAV and Geo-SfM modules favourably in terms of accessibility. After all, a simple search-engine search for *structure from motion photogrammetry tutorial* at the time of submission shows the Geo-SfM module among the top-listed results, and underlines the accessibility factor of the modules in in the practical sense. So too, do the external contributions to both Geo-UAV and Geo-SfM, and the four external participants that independently provided valuable survey feedback to the modules. Both modules also rated positively on clarity, ease of use, diversity of content, and their modern design, though would benefit from being translated into additional languages (improving upon the modules' current *Closed* language rating in the *Open Enough* rubric, Table 2).

Indeed, some of the technologies and software being used were unfamiliar to the students, though this was easily overcome through active facilitation, concise foundational work, and hands-on guidance by instructors. For example, the introduction to the backend, alongside a brief tutorial on how to revise the Jupyter Book files, in particular, cultivated an interest in revising the source materials and update information where it was deemed outdated or inconclusive - a recurring student feedback theme (Table 1). Students easily identified and raised issues, which were then curated and patched by themselves and others, who then also became contributors and co-owners of the content. In addition, the collaborative experience resulted in enhanced collaboration, where multiple student pairs worked together to put more extensive revisions together, including multimedia (e.g., Fig. S2). Students described the practice as increasing their feeling of belonging, with one student reflecting that the ability "to make small changes to the actual site felt inclusive" and another mentioning the benefits of seeing student contributions from past years. Co-creation also led to pedagogic improvements in the resources. Through student-led revisions, the language and content became clearer and better aligned with students' perspectives and level of understanding.

## 4.2 Design choices - lessons learned and future directions

The iterative and open development of educational content demands considerable effort to create an initial environment that is suitable for students to contribute to. This workload is, however, not unlike the creation of other course content such as lecture slides, and, once established, the OERs benefit inherently from remaining accessible and adaptable to future needs with only minimal time required for student-led (decentralised/co-created) revisions.

Indeed, it is encouraging to see that off-the-shelf software and infrastructure now allow for the easy creation, curation, sharing, adaptation, and use of open-source curricula (e.g., Chen and Asta, 2022; Kim et al., 2021; Executable Books Community, 2020). Using Jupyter Book/, changes in course content can be easily tracked and reintegrated where applicable with the source or form the starting point for derived educational content, contributing to the community-driven development of OERs that makes learning more accessible (e.g., Kim et al., 2021). This was particularly useful in the development of Course 2, as we were able to build upon the Geo-SfM module's history tracking and transfer previously removed side-notes on data acquisition to the more appropriate Geo-UAV module. Version control further allows the documentation of changes, and instructors and students alike can easily visualise changes made to the modules over time, and even reinstate previously removed content. Version control also aids in the mitigation of knowledge loss due to e.g. turn-over of faculty staff.

Both the developmental use of such tools, as well as raising awareness of what can be done with them benefits from dedicated tutoring. This was highlighted by students requesting specific feature, such as implementing a search bar, even though search is natively included in the Jupyter Book menubar and students frequently used the menubar in teaching. Other examples include requests on where and how to find educational resources online, which may benefit from having curated (and searchable) portals for thematic content hosted by the community. As a word of caution, student feedback mentioned incoherent cross-linking between different modules as a point of confusion (Table 1) in Course 2, as it was not entirely clear to them which of the two modules required their focus at a given time. We thus note that students benefit from extended introductions to the Jupyter Book interface and backend, even for seemingly obvious functions, as well as from a clear introduction to the structure of the modules at the beginning of a course.

With regards to field teaching, Geo-UAV showcased the benefits of having interactive and portable documentation that can be easily exported and integrated into field-based teaching. Given our and our students' experiences, we are currently developing additional modules that target field instruments (e.g., differential positioning and various geophysical imaging tools) to further investigate the framework's suitability in field teaching. The development (and future implementation) of these modules largely builds upon the key take-aways presented in this study, itemised in Appendix S1. These will also try and find *Open* rather than *Mixed* solutions for the *Open Enough* rubric's *Material costs* and *Assessment* factors. Both are currently *Mixed* due to respectively the use of closed-source softwares and tools (e.g., drones) and the current lack of *Open* forms of assessments that can be taken beyond the classroom activities. This is in part due to the topics covered by Geo-UAV and Geo-SfM and current design choices of the modules, rather than stemming from Jupyter Book framework limitations.

### 4.3 The teachers' perspective

From a teacher's perspective, a key objective of the digital compendiums was to provide lasting, up-to-date course material to a campus with a small department that does not have significant experience nor capacity in developing and maintaining OERs. Another important objective was to create an interactive environment that promotes active learning (Barba et al., 2019; Freeman et al., 2014) and facilitates learning at one's own pace and interest, which are key to learner-centred and asynchronous learning (Georgiadou and Siakas, 2006). Herein the use of GIFs certainly took an important role.

GIFs provided visual and stepwise instructions that greatly simplified otherwise abstract instructions, supplementing the narrative text with easy-to-follow graphics. In addition to their stated learning values, it certainly helped that GIFs can be easily made at low file sizes and feature a low participation barrier for co-creation, as evident from pull requests by students (Fig. S2). The format lends itself exceptionally well for short visual instructions, yet can be easily "overdone" in terms of information density. In case of the latter, we noticed an increase in questions at the cost of independent learning while in class, highlighting the fine balance in its implementation. Further research is thus needed to optimize GIF content for teaching, as previously done for videos (e.g., Guo et al., 2014).

With students actively co-creating and maintaining learning and multimedia resources, we experienced a significant drop in preparatory workload and instead enabled work on more in-depth resources and specific content requested by students. We also observed that this shifted lectures from a teacher-centric to a learner-centric model that revolved around student-led discussions of findings and design choices. Both aspects simultaneously freed up time and allowed instructors to step in only when really needed. As noted from a student's remark, this was greatly appreciated and provided a unique sense of inclusivity and resulted in a hands-on approach that lectures on a similar topic elsewhere had lacked. The asynchronous and hybrid nature of the modules thus seems to have lowered the participation barrier which may also benefit non-traditional learners and students from underrepresented groups who may have less initial experience with either of the topics covered by the modules.

# 5 Conclusion

This study designed and explored students' attitudes towards educational Jupyter Books hosted on the platform. In summary, Jupyter Book modules can be easily created, shared, adapted, remixed, and, importantly, are very user friendly. Quantitative survey responses indicated a positive student perception to the learner-centric learning environment as well as the co-creation possibilities provided by the Jupyter Book/ framework. The interactive multimedia environment was positively experienced by the students and facilitated asynchronous and active learning. It drove engagement, interest, and exploration of concepts that benefitted both learning and scientific thinking. GIFs were also seen as a positive addition, yet work remains to establish optimal playtime durations. The collaborative nature of the modules was instrumental in cultivating an interest in revising the source materials and updating information where it was deemed outdated or unclear, both by students and instructors alike, and regardless of the contributor's background, affiliation or level of experience. We found that co-creation can decrease the workload to maintain and expand up-to-date course content, thus accomplishing one of our key objectives: to provide lasting, up-to-date course material to a campus with a small department that does not have significant experience nor capacity in developing and maintaining OERs. We also found that Project Jupyter tools can be easily adapted to create a learning environment more suitable for co-creation, requiring only minimal former programming experience. These findings, along with students' positive assessment of the Jupyter Book framework's inclusivity, diversity, and accessibility, contribute to the *mostly open* ranking both modules attained within the *Open Enough* framework of ranking open pedagogics openness.

In closing, we hope that by documenting our approach to co-creating OER-P content, we have set an important step in a community-wide effort to catalogue, develop and co-create educational content, and make these openly available and findable to users. However, such an effort can only succeed through an interdisciplinary approach in which scientists, social scientists and students co-create teaching resources and assess course design and learning in parallel.

*Data availability.* The source material for the Geo-UAV and Geo-SfM modules, as well as that of Geo-MOD (Course 2) is freely available from their respective Zenodo repositories, available alongside URLs to the compiled books in Table 3.

**Table 3:** Data availability of the modules, including URL references.

| Module | URL | Reference |
|--------|-----|-----------|
| Geo-MOD | https://unisvalbard..io/Geo-MOD | Betlem et al. (2024) |
| Geo-UAV | https://unisvalbard..io/Geo-UAV | Rodes et al. (2024) |
| Geo-SfM | https://unisvalbard..io/Geo-SfM | Betlem and Rodes (2024) |

*Author contributions.* PB: Conceptualization, Methodology, Software, Validation, Formal analysis, Investigation, Resources, Data Curation, Writing – Original Draft, Writing – Writing & Reviewing, Visualization, Project administration, Funding acquisition, Project administration. NR: Methodology, Software, Investigation, Resources, Writing – Writing & Reviewing, Visualization, Funding acquisition, Project administration. SMC: Resources, Writing – Writing & Reviewing, Funding acquisition, Project administration. MVK: Conceptualization, Methodology, Writing – Original Draft, Writing – Writing & Reviewing, Supervision.

*Competing interests.* The authors declare that they have no conflict of interest.

*Ethical statement.* The data used in this study were collected on a voluntary and anonymous basis. Identification of individual participants in the questionnaire is impossible. The questionnaire was developed with the Norwegian National Ethics Committee's Guidelines for Research Ethics in the Social Sciences and Humanities in mind. Further, the project was internally reviewed through the University Pedagogy Programme at the University Centre in Svalbard, i.e., the host institution.

*Acknowledgements.* Foremost, we thank the participating students for constructive feedback and their eagerness to participate in and co-create the modules. We also thank UNIS colleagues Kim Senger, Aleksandra Smyrak-Sikora, Rafael Horota, and Thomas Birchall for feedback during the first years of implementation. We also acknowledge funding and support from the Norwegian Centre for Integrated Earth Science Education iEarth (Norwegian Agency for International Cooperation and Quality Enhancement in Higher Education grant #101060) and additional funding provided by the Norwegian CCS Research Centre (NCCS; industry and partners and the Research Council of Norway grant #257579). Similarly, we keenly acknowledge the close collaboration with the Svalbox project (co-financed by the University of the Arctic, the Research Council of Norway and the University Centre in Svalbard). We appreciate the academic licenses of Metashape provided by Agisoft, as well as the UAVs and other hardware made available through the Svalbox project. We also thank Henrikki Tenkanen and Vuokko Heikinheimo

and their Automating GIS-processes documentation for introducing us to the world of Jupyter (educational) Books and open

course documentation. Finally, we sincerely appreciate the constructive discussion with reviewers Enze Chen and Jonathan W. Rheinlænder, as well as the excellent editorial handling and constructive feedback by Mathew Stiller-Reeve.

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

**Supplementary Material S1: Do's and don'ts for implementing the Jupyter Book/ framework**

The below provides a brief cheat-sheet for implementing Jupyter Book/ as a teaching platform, mostly targeting narrative content and summarising some of our key experiences and learnings.

– **Read and share the docs**: The Executable Books (Executable Books Community, 2020) project provides extensive documentation for both Jupyter Book and the MyST Markdown language used to write the books. The documentation includes a start-up guide, as well as easy-to-follow topic guides written in simple language. Do not forget to share this with your students for inspiration.

– **Create a minimal working book**: Generate an outline of the to-be-covered topics and create a separate chapter (i.e., one or multiple files) for each and populate the chapter pages with the minimum educational material that needs to be covered in class.

– **Keep it simple**: Going back and forth between different sections (and modules), was shown to confuse students, as was the (attempted) inclusion of too many topics at once. First, try to avoid extensive cross-linking between pages and content blocks and instead design the module to follow a single red thread. Second, rather create supplementary books covering related topics than including too much content at once.

– **Provide examples**: Both narrative and multimedia content should be included in the minimal working book, as well as computational content when applicable. The overlapping multimedia approach provides diverse and asynchronous learning options, in addition to providing a quick lookup sheet for student to adapt source-code snippets from during co-creation.

– **Familiarise students with the framework**: Do not expect students to create content out of thin air. First, students need to be comfortable using the tools and be given ample opportunity and freedom to revise content. This means one must first lay the foundation for co-creation. For example, start with the basics by explaining students how to navigate the Jupyter Book pages, provide a basic introduction on how to use the backend. A simple "hello world" post on is an easy start. Then extend their co-creation skills by introducing more extensive revisions through forks and pull requests, for example asking students to fix spelling mistakes or replace a figure. Another example, taken from the Geo-SfM module, is to ask students to share their results by updating a built-in gallery, in Geo-SfM done by pull-requesting a model tag into a configuration file on the "Uploaded examples" page. Remember, for those without a programming background, such a revision may already feel like extensive programming and quite the achievement.

– **Co-creation over time**: Do not expect pages worth of content to be added by students at once, rather, the minimal working book will evolve over time as revisions and additions culminate in a compendium co-shaped by students.

– **Encourage additions and revisions**: Faster-paced students, those who have taken similar courses elsewhere, or those interested in more advanced scenarios may be eager to extend the course content. This is best done by giving them a well-defined task, which can be as simple as asking them to e.g., document (both text and GIFs) what function X does in program Y or to expand a pre-existing section.

- **Usability vs. functionality**: Use open and/or pre-installed softwares, such as the snipping tool, that are easy to use by both instructors and students alike, rather than overly-complex softwares and tools. These typically make for straight-forward tools that capture content in sufficient quality to be included in the course while being time- and resource-efficient.

- **Keep it concise**: Describe things stepwise and to the point. Try to include only one step per accompanying GIF at a time, opting rather for several than for one long animation.

- **Learning first and foremost**: The students' main focus should be on absorbing and shaping course content, and not on dealing with compilation errors and software bugs. Thus, it is highly advised that instructors maintain control over the "build" process of the Jupyter Book pages. This also allows instructors to inspect changes prior to publishing. Secondly, it is advised to only sporadically re-build the books from their source, ideally when students are not using the resource. This to prevent confusion due to e.g. mismatching pages and unexpected changes.

- **Disseminate**: The open sharing and listing of Jupyter Books (for example in the Jupyter Book Gallery) helps others find, access, integrate and reuse their resources. External collaborators may even contribute to the Jupyter Book, supporting co-creation and collaboration within the greater community.

# Supplementary Material S2: Supplementary Tables and Figures

**Table S1:** Multimedia counts and playtime statistics.

| Module | Feature type | Feature count | Internal/External | Playtime (min) | Playtime (mean) | Playtime (max) | Playtime (std) |
|--------|--------------|---------------|-------------------|----------------|------------------|-----------------|-----------------|
| Geo-SfM | Animated GIFs | 17 | 17/0 | 8.4 s | 23.7 s | 78.0 s | 17.9 s |
| Geo-UAV | Animated GIFs | 14 | 14/0 | 3.8 s | 8.1 s | 13.0 s | 2.3 s |
| Geo-SfM | Video | 4 | 1/3 | 130 s | 171.8 s | 206 s | 32.9 s |
| Geo-UAV | Video | 8 | 2/6 | 39 s | 178.6 s | 388 s | 101.4 s |

**Table S2:** Feedback on the average number of times an animation or video was replayed and paused.

| Frequency | Ani. rewatch | Vid. rewatch | Vid. pause |
|---|---|---|---|
| 0 | 6 | 18 | 18 |
| 1-3 | 21 | 23 | 17 |
| 4-6 | 12 | 0 | 4 |
| 7-10 | 2 | 0 | 2 |

**Table S3:** Questions and answer options.

| Question/statement | Answers | Ref. |
|---|---|---|
| **BACKGROUND INFORMATION** | | |
| I am affiliated/enrolled with UNIS... | YES \| NO | |
| My (educational/scientific) background mostly corresponds to....Biology | YES \| NO | |
| My (educational/scientific) background mostly corresponds to....Geology | YES \| NO | |
| My (educational/scientific) background mostly corresponds to....Geophysics | YES \| NO | |
| My (educational/scientific) background mostly corresponds to....Technology | YES \| NO | |
| My (educational/scientific) background mostly corresponds to....Safety | YES \| NO | |
| My (educational/scientific) background mostly corresponds to....Guiding | YES \| NO | |
| My (educational/scientific) background mostly corresponds to....Other | YES \| NO | |
| | | |
| **BACKGROUND KNOWLEDGE** | | |
| I was familiar with programming (e.g., Python, R, matlab) | − \| - \| -+ \| + \| ++ | Fig. S1 |
| I was familiar with Jupyter Notebook/Lab | − \| - \| -+ \| + \| ++ | Fig. S1 |
| I was familiar with Jupyter Book/Executable Book Project/Sphinx/Read The Docs | − \| - \| -+ \| + \| ++ | Fig. S1 |
| I was familiar with YouTube | − \| - \| -+ \| + \| ++ | Fig. S1 |
| I was familiar with animated GIFs | − \| - \| -+ \| + \| ++ | Fig. S1 |
| I was familiar with git (e.g., GitHub) | − \| - \| -+ \| + \| ++ | Fig. S1 |
| | | |
| **DEVICE USED** | | |
| I used the following device to read/interact with the compendiums:.Mobile phone | YES \| NO | |
| I used the following device to read/interact with the compendiums:.Tablet | YES \| NO | |
| I used the following device to read/interact with the compendiums:.Desktop | YES \| NO | |
| | | |
| **RATINGS AND OBJECTIVES** | | |
| What were your (learning) objectives when following the XXX Compendium? | Open | |
| Future documentations - your compendium ideas? | Open | |
| | | |
| Rate your overall experience of using the XXX Compendium on a scale from 0 to 10, with 0 being extremely dissatisfied and 10 being extremely satisfied. | 0 to 10 | |
| Rate your overall experience of learning with the XXX Compendium on a scale from 0 to 10, with 0 being extremely dissatisfied and 10 being extremely satisfied. | 0 to 10 | |
| | | |
| **COMPENDIUM INFORMATION** | | |
| The XXX Compendium met my needs | − \| - \| -+ \| + \| ++ | Fig 2 |
| The topics covered by the XXX Compendium were relevant to the course | − \| - \| -+ \| + \| ++ | Fig 2 |
| The XXX Compendium pages were easy to navigate | − \| - \| -+ \| + \| ++ | Fig 2 |
| The XXX Compendium content was clear | − \| - \| -+ \| + \| ++ | Fig 2 |
| The XXX Compendium content was too difficult | − \| - \| -+ \| + \| ++ | Fig 2 |
| I enjoyed using the XXX Compendium | − \| - \| -+ \| + \| ++ | Fig 2 |
| I would recommend the XXX Compendium to others | − \| - \| -+ \| + \| ++ | Fig 2 |
| I will use/have used the XXX Compendium after the module ended | − \| - \| -+ \| + \| ++ | Fig 2 |
| | | |
| What did you like most about the XXX Compendium? Try to come up with at least two examples. | Open | Table 1 |
| What did you like least about the XXX Compendium? Try to come up with at least two examples. | Open | Table 1 |
| | | |
| **MULTIMEDIA QUESTIONS** | − \| - \| -+ \| + \| ++ | Fig. 3 |
| The animated GIFs and videos explained the content clearly | − \| - \| -+ \| + \| ++ | Fig. 3 |
| The animated GIFs and videos supplemented and clarified the text | − \| - \| -+ \| + \| ++ | Fig. 3 |

| Question/statement | Answers | Ref. |
|---|---|---|
| The use of animated GIFs and videos helped me better understand the material | – \| - \| -+ \| + \| ++ | Fig. 3 |
| The quality of the animated GIFs was high | – \| - \| -+ \| + \| ++ | Fig. 3 |
| The quality of the videos was high | – \| - \| -+ \| + \| ++ | Fig. 3 |
| The animated GIFs played too fast | – \| - \| -+ \| + \| ++ | Fig. 3 |
| The videos played too fast | – \| - \| -+ \| + \| ++ | Fig. 3 |
| The animated GIFs lasted too long | – \| - \| -+ \| + \| ++ | Fig. 3 |
| The videos lasted too long | – \| - \| -+ \| + \| ++ | Fig. 3 |
| Being able to pause the animated GIFs would be useful | – \| - \| -+ \| + \| ++ | Fig. 3 |
| A voice-over/spoken instruction would make the animations and videos better to understand | – \| - \| -+ \| + \| ++ | Fig. 3 |
| I typically watched the videos at faster playback speeds | – \| - \| -+ \| + \| ++ | Fig. 3 |
| I typically watched the videos at slower playback speeds | – \| - \| -+ \| + \| ++ | Fig. 3 |
| | | |
| On average, I rewatched individual animated GIFs x times | 0 \| 1 - 3 \| 4 - 6 \| 7 - 10 | Table S2 |
| On average, I rewatched individual YouTube videos x times | 0 \| 1 - 3 \| 4 - 6 \| 7 - 10 | Table S2 |
| On average, I paused YouTube videos x times | 0 \| 1 - 3 \| 4 - 6 \| 7 - 10 | Table S2 |
| | | |
| What did you like about the mixed use of animated GIFs and videos in the compendiums? What worked? | Open | Table 1 |
| What did you dislike about the mixed use of animated GIFs and videos in the compendiums? What did not work? | Open | Table 1 |
| | | |
| DID YOU KNOW? | | |
| Individual compendium pages can be generated as PDFs from within the pages themselves? | YES \| NO | |
| You can contribute and suggest changes directly from the compendium pages? | YES \| NO | |
| The raw material for each compendium is openly available on GitHub? | YES \| NO | |
| GitHub hosts an issue and bug tracker for each of the compendiums? | YES \| NO | |

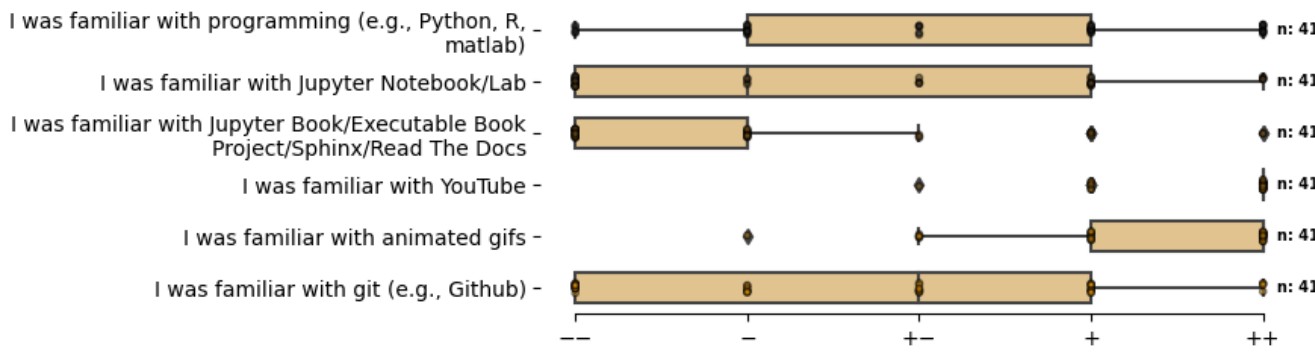

**Figure S1.** Assessment of prior knowledge/experience to the implemented digital tool sets one which the compendiums are built.

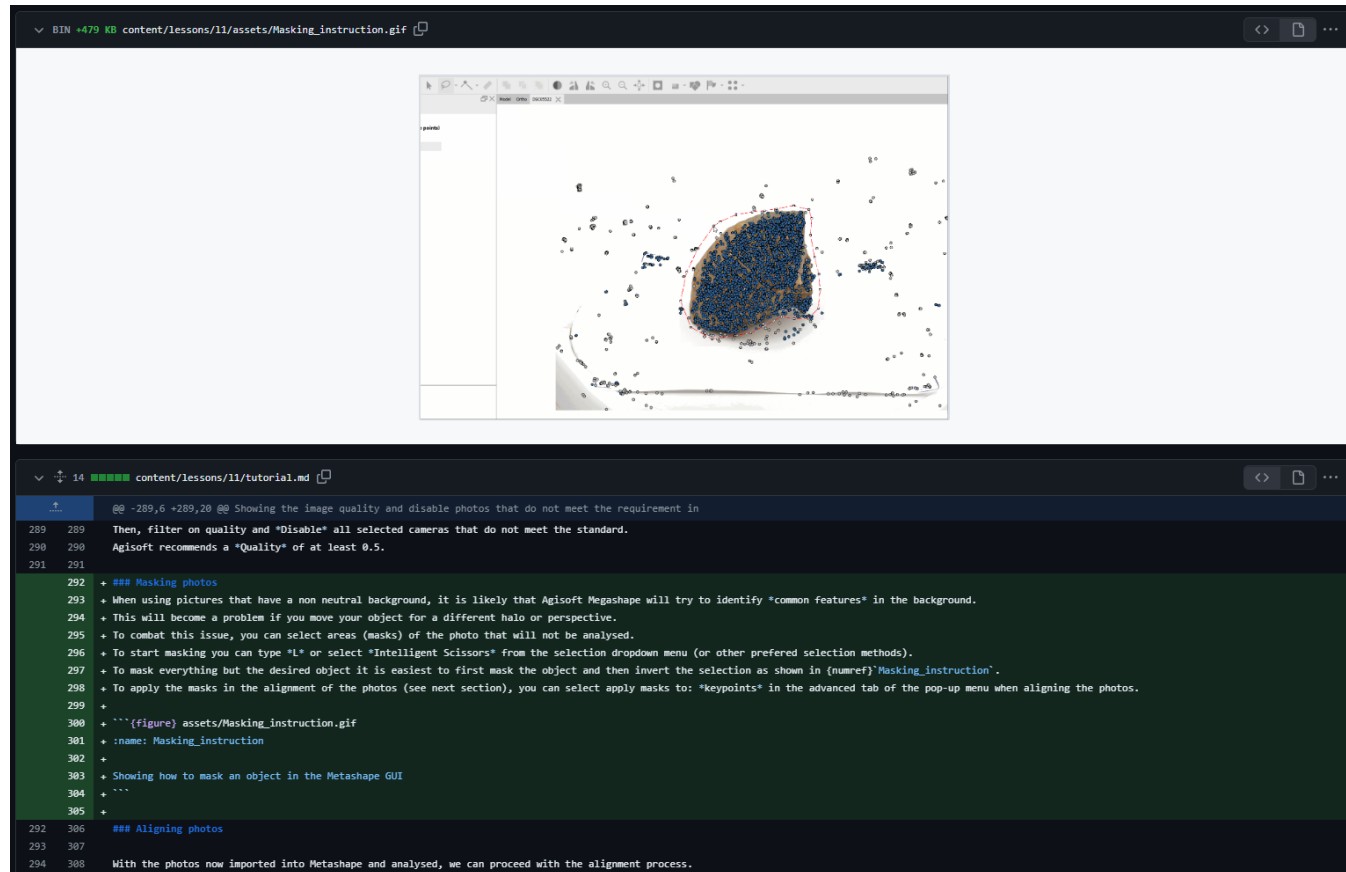

**Figure S2.** Student contributions ranged from single edits and suggestions, to multi-paragraph revisions and newly-recorded animations. Shown here is the student-contributed revision that documents the masking of photos in Agisoft Metashape and added it to the Geo-SfM tutorial (lesson 1). Note that the contribution is formatted in MyST Markdown and includes both text, an image code-block, and the self-recorded animation. Pull request link: https://github.com/UNISvalbard/Geo-SfM/pull/66.