# Peer review of "Jupyter Book as an open online teaching environment in the geosciences: Lessons learned from Geo-SfM and Geo-UAV"

_Geoscience Communication, 2024_

## Author Comment (AC1)

**1  Introduction**

The concept of openness and sharing has become a core value and commitment across many disciplines and fields .The open-source and FAIR (findable, accessible, interoperable, reusable) data (Wilkinson et al., 2016) stewardship movements share common principles with in relation to both teaching and research. With teaching specifically, open pedagogy (OP) (Rocca-Serra et al., 2023; Wiley and Hilton, 2018). Through open-source tools, FAIR data, and open educational materials, OP provides envisions a more democratised, accessible, and affordable learning environment wherein neither students nor educators are bound by expensive software licences, proprietary data, or the limited perspectives or costs of individual textbooks (Abernathy, 2023). Specifically, (Abernathy, 2023; Wiley and Hilton, 2018; McNally and Christiansen, 2019;

20

Harrison et al., 2022; Matkin, 2009). OP is an educational approach that emphasizes transparency, collaboration, student-driven learning, and the use of open educational resource resources (OERs) (Hegarty, 2015; Wiley and Hilton, 2018). Specifically, it is defined as any type of educational teaching that is in the public domain or accessible with an open licence (Audrey Azoulay, 2019). Unlike conventional, proprietary educational materials and practices,

Despite increasing adoption, OP remains far from a formalised and recognised standard, but rather a loose set of aspirational guidelines that are difficult to navigate and interpret as a whole (Christiansen and McNally, 2022; Tietjen and Asino, 2021; Weller, 2014; Wiley and Hilton, 2018). At its core, OP encourages educators and students to actively engage in the creation, adaptation, and sharing of educational materials, rather than relying on conventional, proprietary educational materials and practices. In so doing, it encourages OP facilitates transparency in teaching practices and makes learning materials openly accessible to a broader audience, enhancing the visibility of educational content and allowing for wider participation (Jhangiani and Biswas-Diener, 2017).

Despite the increasingly wide adoption, OP remains far from a formalised and recognised standard, but rather a loose set of aspirational guidelines that are "essentially impossible" to reconcile (Wiley and Hilton, 2018; Tietjen and Asino, 2021; Christiansen and McNally, 2022; Weller, 2014). (McNally and Christiansen, 2019) suggest OP openness can be evaluated based on the eight primary factors, including copyright, accessibility, language, support costs, assessment, digital distribution, file format, and cultural considerations. None of these are binary "open", underlining the difficulty of defining what is and is not open (McNally and Christiansen, 2019). The OER-enabled pedagogy (OER-P) subset of OP implements many of these factors and is governed by a set of five specific rights, the so-called 5 Rs of OER that regulate openness and reduce the problem of disposable assignment (Wiley, 2013). These consist of the right to

**1.1 Open Educational Resources in pedagogy**

The creation and use of OERs is integral to OP. OERs, being any type of teaching material that is in the public domain or accessible with an open licence, are best examined through the 5Rs (retain, reuse, revise, remix and redistribute educational content (Tietjen and Asino, 2021; Wiley, 2013; Wiley and Hilton, 2018). , redistribute) (Audrey Azoulay, 2019; Wiley, 2013). They serve as both a critique and alternative to conventional scholarly and educational publishing and have seen an uptake in recent years (Tietjen and Asino, 2021; Wiley, 2013; Wiley and Hilton, 2018).

[revised manuscript text omitted]

100 of the environment and assess ), assess user and student learning experiencesof using and contributing to the course modules, and appraise the co-creation possibilities.

feedback solutions. The students were then asked to work through the course modules in pairs, applying the concepts of pair learning to further enhance collaborative learning (Nagappan et al., 2003; Drey et al., 2022).

In the Geo-SfM module, students were invited to "catalogue" their processing results by voluntarily submitting their work to a gallery page. This was done either in collaboration with instructors or individually through a pull request to a configuration file. Required metadata included a link to the results, the course year and course ID that, once compiled, provided an example gallery and overview of past work.

The GitHub platform, including its Classroom tools, has previously been shown to improve the educational experience for students and teachers (e.g., Zagalsky et al., 2015; Fiksel et al., 2019), and facilitates open hosting of documentation. The use of

150

155

**4    Discussion**

Unlike proprietary lecture materials and technologies, the entry barriers to entry for students learning with open-source resources such as Jupyter Book can be very low (Barba et al., 2019). For many of the students in our courses, the Geo-UAV and Geo-SfM modules were their first foray into the large and growing ecosystem of such tools. Like open-source software (Khan and Ur Rehman, 2012), OERs have the unique opportunity to deliver inherently collaborative, transparent workspaces that extend beyond the original authoring institution or idea (Caswell et al., 2008).

The present study explored students' perceptions of two Jupyter Book-based modules that were designed with the explicit goals to increase openness, diversity, and student co-creation in creating OERs in OP. Certainly, OERs are hardly new to the academy; however, what could or should "count" as OERs has become a source of concern for scholars and advocates who note the casual use of the term "open" for materials that neglect or obstruct the 5Rs (typically because of copyright restrictions) (Wiley and Hilton, 2018). It can be useful then to consider how "openness' can be understood and assessed. McNally and Christiansen (2019) suggest OER openness can be evaluated based on the eight primary factors, including copyright, accessibility, language, support costs, assessment, digital distribution, file format, and cultural considerations. They experiment with using these criteria with a three-part scale (closed, mixed, most open) - meaning that resources may be "most open" with regard to some criteria and "mixed" or "closed" in relation to others. Their work suggests that the relative openness of OERs (conceptualized through the 5Rs) can and should be evaluated by educators, as we have done here.

In the discussion that follows, we use students' survey responses to assess these evaluate relative openness/accessibility and other pedagogical factors and summarise our findings grade the two course modules accordingly through an *Open Enough* rubric (Christiansen and McNally, 2022) (Table 2; treating *Harvestability* as a *Technical* rather than *Pedagogical* factor). Both Geo-UAV and Geo-SfM (and the Jupyter Book/GitHub framework as a whole) rank high on openness, outranking many of those rated by Christiansen and McNally (2022). Key considerations are the modules' learner-centred design and the implementation of collaborative design choices.

Perhaps the most important reflections came on the use and integration of animations students' reflections were on the integration of GIFs
and videos in addition to the rich text descriptions, which were stated to greatly benefit the diversity and accessibility of the
course content. Where shorter animations GIFs of up to a few seconds were preferred to explain single steps, students seemed to
prefer pausable videos for content with longer playtimes that covered multistep processes, similar to observations made in
educational video design studies (e.g., Guo et al., 2014). During plenum discussions, students largely agreed with our hypothesis also
largely agreed that videos form a higher participation-barrier for co-creation, especially given the ease with which short ani-
mations can be re-recorded and updated, and higher cost of videos in terms of time, IT skills, and storage requirements. Thus, in addition to being low-bandwidth,
animated gifs were found to be ideally suited as long as the content was sufficiently decomposed into digestible chunks as seen through student contributions
(Fig. A1). The consecutive use of single-step animations acted as a form of signaling, or cueing, of different parts of the
greater process, which has been shown to help direct learner attention and improves knowledge retention and transfer
(de Koning et al., 2009; Mayer and Moreno, 2003; Brame, 2015). Further studies are, however, needed to ascertain these
findings and find optimal playtime durations for animated and video content the animated GIF content, as has previously been done
for educational videos (e.g., Guo et al., 2014).

Indeed, some of the technologies and software being used were nascent and unfamiliar to students, though this was easily

Book/GitHub framework. These lay the groundwork for future activities that are needed to quantify student perceptions of these and other aspects, which we only briefly touched upon in the current study. Still, our results provide valuable insight into how to design and co-create future OER-P content.

From the perspective of instructors, we are excited As instructors, it is encouraging to see that open-source off-the-shelf software and infrastructure has matured to have reached the point where open-source curricula can be easily created, shared, adapted, and, importantly, used and found . Like us and our students, other educators (e.g., Chen and Asta, 2022; Kim et al., 2021; Executable Books Community, 2020). With these tools, instructors and learners alike have access to and can remix different compendium versions and works for their course-specific needs. These (learning) needs. Moreover, using for example the GitHub backend, adaptations can be easily tracked through the GitHub backend and reintegrated where applicable with the source, alternatively form the foundation for derived educational content. Indeed, such adaptations often find their way back to the original modules and contribute to a in doing so, such adaptations and co-creations contribute to the community-driven development of OERs that makes learning more accessible (e.g., Kim et al., 2021).

At the time of submission, a simple search-engine search for *structure from motion photogrammetry tutorial* shows the Geoshould be done with caution, as reflected on in Course 2 evaluations by students. Extensive cross-linking between the Geo-UAV and Geo-SfM modules was often mentioned as a point of confusion, and it may thus be better to integrate, rather than link, corresponding materials in the correct pedagogical structure.

Course 2 also illustrated that the chosen JupyterBook /GitHub framework worked well for both in- and outdoor settings. The Geo-UAV module with its field days, in particular, showcased the possibility of having interactive and portable documentation that can easily be taken into the field and integrated into field-based teaching. Given this success, we are planning on developing additional modules that target field instruments such as differential positioning and various geophysical imaging tools, some of which are already available through a dedicated module portal It is also important to note that students benefited from extended introductions to the JupyterBook interface. This became clear to us only after specific feature requests such as the implementation of a search bar were made, even though search and a search bar is automatically included in the JupyterBook menubar. Here it sits next to buttons used for raising issues and generating portable PDF documents, which students frequently used to contribute and when exporting notes to bring along in the field.

With students actively co-creating and maintaining aiding the maintenance of learning resources, we experienced a significant drop in preparatory workload and instead enabled work on more in-depth resources and specific content requested by students.

435 *Data availability.* The source material for the Geo-UAV and Geo-SfM modules, as well as that of Geo-MOD (Course 2) is freely available from their respective Zenodo repositories, available alongside URLs to the compiled books in Table 3.

Table 3. Data availability of the modules, including URL references.

| Module | URL | Reference |
|---|---|---|
| Geo-MOD | https://unisvalbard.github.io/Geo-MOD | Betlem et al. (2024) |
| Geo-UAV | https://unisvalbard.github.io/Geo-UAV | Rodes et al. (2024) |
| Geo-SfM | https://unisvalbard.github.io/Geo-SfM | Betlem and Rodes (2024) |

[Figure]

Figure A1. Student contributions ranged from single edits and suggestions, to multi-paragraph revisions and newly-recorded animations. Shown here is the student-contributed revision that documents the masking of photos in Agisoft Metashape and added it to the Geo-SfM tutorial (lesson 1). Note that the contribution is formatted in MyST Markdown and includes both text, an image code-block, and the self-recorded animation. Pull request link: https://github.com/UNISvalbard/Geo-SfM/pull/66.

---

## Author Comment (AC2)

**Jupyter Book as an open online teaching environment in the geosciences: Lessons learned from Geo-SfM and Geo-UAV**

comprised both field and classroom teaching and were iteratively revised over a four-year period. The modules covered the acquisition of drone data and subsequent processing of 3D modelsand were iteratively revised over a four-year period. Each module

**1 Introduction**

The concept of openness and sharing has become a core value and commitment across many disciplines and fields .The open-source and FAIR (findable, accessible, interoperable, reusable) data (Wilkinson et al., 2016) stewardship movements share common principles with in relation to both teaching

20 and research. With teaching specifically, open pedagogy (OP) (Rocca-Serra et al., 2023; Wiley and Hilton, 2018). Through open-source tools, FAIR data, and open educational materials, OP provides envisions a more democratised, accessible, and affordable learning environment wherein neither students nor educators are bound by expensive software licences, proprietary data, or the limited perspectives or costs of individual textbooks (Abernathy, 2023). Specifically, (Abernathy, 2023; Wiley and Hilton, 2018; McNally and Christiansen, 2019;

Harrison et al., 2022; Matkin, 2009). OP is an educational approach that emphasizes transparency, collaboration, student-driven learning, and the use of open educational resource resources (OERs) (Hegarty, 2015; Wiley and Hilton, 2018). Specifically, it is defined as any type of educational teaching that is in the public domain or accessible with an open licence (Audrey Azoulay, 2019). Unlike conventional, proprietary educational materials and practices,

Despite increasing adoption, OP remains far from a formalised and recognised standard, but rather a loose set of aspirational guidelines that are difficult to navigate and interpret as a whole (Christiansen and McNally, 2022; Tietjen and Asino, 2021; Weller, 2014; Wiley and Hilton, 2018). At its core, OP encourages educators and students to actively engage in the creation, adaptation, and sharing of educational materials, rather than relying on conventional, proprietary educational materials and practices. In so doing, it encourages OP facilitates transparency in teaching practices and makes learning materials openly accessible to a broader audience, enhancing the visibility of educational content and allowing for wider participation (Jhangiani and Biswas-Diener, 2017).

Despite the increasingly wide adoption, OP remains far from a formalised and recognised standard, but rather a loose set of aspirational guidelines that are "essentially impossible" to reconcile (Wiley and Hilton, 2018; Tietjen and Asino, 2021; Christiansen and McNally, 2022; Weller, 2014). (McNally and Christiansen, 2019) suggest OPopenness can be evaluated based on the eight primary factors, including copyright, accessibility, language, support costs, assessment, digital distribution, file format, and cultural considerations. None of these are binary "open", underlining the difficulty of defining what is and is not open (McNally and Christiansen, 2019). The OER-enabled pedagogy (OER-P) subset of OP implements many of these factors and is governed by a set of five specific rights, the so-called 5 Rs of OER that regulate openness and reduce the problem of disposable assignment (Wiley, 2013). These consist of the right to

**1.1 Open Educational Resources in pedagogy**

The creation and use of OERs is integral to OP. OERs, being any type of teaching material that is in the public domain or accessible with an open licence, are best examined through the 5Rs (retain, reuse, revise, remixand redistributeeducational content (Tietjen and Asino, 2021; Wiley, 2013; Wiley and Hilton, 2018). , redistribute) (Audrey Azoulay, 2019). They serve as both a critique and alternative to conventional scholarly and educational publishing and have seen an uptake in recent years (Tietjen and Asino, 2021; Wiley, 2013; Wiley and Hilton, 2018).

OER-Ps can be seen as an extension of the knowledge-building framework, which values students' work primarily for what it contributes to the community, and secondarily for what it reveals about individual students' knowledge (Bereiter and Scardamalia, 2014; Tietjen and Asino, 2021). After all, having the right to freely distribute materials with the broader outside world inherently increases the value of the work (Wiley, 2013), and it is this key element that sets Wiley and Hilton (2018) propose the use of OER-enabled pedagogy (OER-Papart from other forms of OP and teaching practices, whilst still benefiting from the OP framework (Andrade et al., 2011)) as way of conceptualizing what can be pedagogically possible through use of the 5Rs of OER.

Open distribution and access further saves money and reduces cost, for instance, by minimising duplication and the generation of disposable material, and extend the usability of resources (Wiley and Hilton, 2018). As a subset of OP, OER-P benefits from the participatory nature of OP while acknowledging the role

Asino, 2021): Where small departments or single lecturers with little experience in online teaching may struggle to hybridise a class, a community of (networked) OER-P practitioners with complementing expertises have far better chances to update and revise educational materials and courses, especially when aided by student-led co-creation. Co-creation by pursued in collaboration with students.

**1.2 OER-P and co-cration with students**

Importantly, co-creation of OERs and through OER-P practices with students has the benefit of increasing diversity in teaching materials, thereby enhancing engagement and improving learning outcomes of individuals who are otherwise underrepresented in education (Biddle and Clinton-Lisell, 2023; Lambert, 2018; Kelly et al., 2022; Nusbaum, 2020). Overlapping with scholarship on Students as Partners, OER-P strategies enable what Bovill and Woolmer (2019) describe as co-creation *in* curriculum and co-creation *of* curriculum. Significantly, co-creation in/of the curriculum strategies are clearly noted for their potential to redress power relations between teachers and students, reframe knowledge and knowledge production and "counter the increasing commodification of learning" (Bovill and Woolmer, 2019, p. 408). It is from this point, that our project emerges - we are curious about the potential of OER resource development as a transformative pedagogical practice that is undertaken collaboratively with students.

**1.3 Tools for OER development**

2020), akin a user-editable and annotatable "unbook" that is not subject to the dramatic inflation of traditional textbooks (Woodworth, 2011; Harrison et al., 2022; Matkin, 2009)and . Simply put, Jupyter Book provides an interface for building publication-ready books ("Jupyter Books") from computational (e.g., Jupyter Notebooks) and narrative (e.g., text files, images) content (Executable Books Community, 2020). The environment can be easily integrated with co-creation and version/source control solutions such as git. With a strong focus on the collaborative development, creation, and expansion of documentation, along with open licencing options, we decided to test whether Jupyter Books can indeed act as an a diverse, equitable and inclusive learning environment that embraces the three pillars of "open" social justice (i.e., redistributive, recognitive, and representational) described by Lambert (2018) and Biddle and Clinton-Lisell (2023).

This article documents In this contribution, we first document the implementation of Jupyter Books Book and GitHub in the design of two geoscience undergraduate modules on data acquisition and processing as part of a transition to OER-P teaching. The two integrated, interactive online

textbooks cover and detail best practices in the acquisition and processing of unmanned aerial vehicle (UAV) -based data (Geo-UAV) and the subsequent multi-view stereo (MVS) data acquisition and structure-from-motion (SfM) photogrammetry processing(Geo-SfM). Our design was informed and inspired by existing textbooks and tutorials published using Sphinx and Jupyter Book (Henrikki Tenkanen et al., 2023, 2022; Lehmann, 2011; Executable Books Community, 2020; Rhoads and Gan, 2022) that showcases the ease of integrating interactive components within narrative course content. Animations and animated gifs are an important design-choice to increase engagement and lower the barriers for participation (Bakhshi et al., 2016). Because of their capacity to capture short animations, and generally small file sizes, gifs have become a key communication tool on par with other visual media (Miltner and Highfield, 2017; Bakhshi et al., 2016). The adoption of gifs for commercial purposes illustrates the adaptability of the format, and gifs are increasingly used to illustrate points, provide information, advertise, and even augment news and information (Miltner and Highfield, 2017). It is thus not surprising that gifs have been previously used in educational settings (e.g., Altintas et al., 2017; Talati et al., 2020; Russell, 1999; Brisbourne et al., 2002). In this contribution, we apply these and other pedagogical learnings and discusses the implementation of the Jupyter Book environment, including the integrated , respectively, as part of a transition to OER-P teaching at a small campus. We then demonstrate the openness and accessibility of the framework (aided by the use of animationsand traditional course content, as a tool for enhanced geoscientific learning. We demonstrate the applicability of the environment and assess ), assess user and student learning experiencesof using and contributing to the course modules, and appraise the co-creation possibilities.

**2 Methods and data**

**2.1 Context and participants**

UAV module was developed as a teaching-aid based on experiences and best-practices acquired through the Svalbox project and its spin-offs (Senger et al., 2021; Betlem et al., 2023). As fieldwork forms a large component of data acquisition, our study implicitly tested the portability of Geo-UAV into field-based teaching, either by accessing the tutorial online in the field, or by exporting PDF pages prior to heading out.

**2.2 Module and course design**

Our design was informed and inspired by existing textbooks and tutorials published using Sphinx and Jupyter Book (Henrikki Tenkanen et al., 2023, 2022; Lehmann, 2011; Executable Books Community, 2020; Rhoads and Gan, 2022) each detailing the ease of integrating interactive components and narrative course content. The Geo-SfM and Geo-UAV modules modules were designed to facilitate a learning environment that is inclusive, accessible, and diverse in terms of representation of information, learning styles and perspectives. The Jupyter Book environment was implemented as the

160

presentations replaced graded assessments and exams. Classroom teaching further implemented the colloquial sharing of results and experiences during daily recaps in which students presented both their results and stumbles, with feedback and possible solutions mostly provided by other working groups. Peer-to-peer evaluation was also encouraged for pull-requests and revisions suggested to the courses, though were not part of the grading process. The setup of the modules,

165    implementing gradual and asynchronous learning, naturally facilitated grading through module completion and participation. In Course 2, the shared assessment for the individual sessions was certified and documented in a course certificate, listing the accomplished learning objectives, and stating their equivalent.

Both GIFs, given their capacity to capture short animations and generally small file sizes, have become a key communication tool on par with other visual media (Bakhshi et al., 2016; Miltner and Highfield, 2017) and their inclusion has been shown

170 to increase engagement and lower the barriers for participation (Bakhshi et al., 2016). For this reason, we implemented both shorter and longer animations and videos were implemented in addition to to supplement videos, detailed plain-language summaries and static figures in order to improve the accessibility of learning materials. Specifically, we were interested in determining

Book and GitHub platforms. Fig. 2, Fig. 3 list several questions and statements from the survey along with quantified student feedback. Students' questionnaire feedback was categorised and coded as either *constructive criticism* or *Positive feedback* with examples provided alongside.

Implementing feedback from the 2023 courses, we provided the class of 2024 was provided (Geo-SfM module) with a more extensive, preparatory three-hour tutorial on how to contribute through so-called forks and pull-requests. Forks and pull-requests As these tools allow more sophisticated changes to be made to the content pages but require a (documented) review by other participants and course instructors (source code of the) pages, a review and approval by others is needed prior to integration into the live module pages. Herein each students were given a prominent role to review one-another's proposed revisions and additions before final approval by instructors. As each pull-request interaction is documentedand , the process automatically attributes co-creators . Assessment of the class of 2024 thus in addition focussed on what can be done to lower the barrier to to the revised resource. Naturally, the focus of our pedagogics study broadened slightly to understand what the use of these tools meant in terms of lowering the barrier for co-creation within the implemented Jupyter Book/GitHub framework, as well as its apparent value to participantstheir apparent learning value to the students.

and how they perceived the multimedia use in the modules. Open-ended remarks (Table 1) on the use of animations and videos within the modules resonated well with the quantitative feedback (Fig. 3) given by the students.

Perhaps the most important reflections came on the use and integration of animations and videos in addition to the rich text descriptions, which were stated to greatly benefit the diversity and accessibility of the course content. Overall, students reported agreement that the use of animations and videos greatly supplemented the main text, and that the quality of animations and videos was high. Indeed, students also indicated that the playtime of multi-step animations (GIFs), in particular, was long and that a pause-function would have been a welcome addition (Fig. 3). This was in agreement with open-ended responses, such as that students did not like having "to wait for the loop to end to see again the info [they] wanted to see" and that they "had to play it [GIFs] several times to identify all steps", which is further evident from self-reported playtime statistics (Table S2). That said, students largely agreed that the use of GIFs was "useful to assist with processes and to reduce the amount of learning through reading".

270 For both modules, a selection of students reflected on a perceived information disparity between the main body text, multimedia elements, and instructions. This included occurrences of (outdated) animations that were recorded for a previous version of the software, content displayed in multimedia but not the main text body (and vice versa), and the extent of operations covered by the modules versus more advanced usage The ease with which students recorded and updated short animations (Fig. S2) point us to the hypothesis that GIFs form a lower participation barrier for cocreation than videos, which is partly supported by our findings. Important work, however, remains to be done to ascertain optimal implementations and other

275 benefits of educational GIF content, as previously done for educational videos (e.g., Guo et al., 2014).

**4 Discussion**

290

Unlike proprietary lecture materials and technologies, the entry barriers to entry for students learning with open-source resources such as Jupyter Book can be very low (Barba et al., 2019). For many of the students in our courses, the Geo-UAV and Geo-SfM modules were their first foray formed their first introduction into the large and growing ecosystem of such tools. Like open-source software (Khan and Ur Rehman, 2012), OERs have the unique opportunity to deliver inherently collaborative,

295 transparent workspaces that extend beyond the original authoring institution or idea (Caswell et al., 2008).

The present study explored students' perceptions of two Jupyter Book-based modules that were designed with the explicit goals to increase openness, diversity, and student co-creation in creating OERs in OP. Certainly, OERs are hardly new to the academy; however, what could or should "count" as OERs has become a source of concern for scholars and advocates who note the casual use of the term "open" for materials that neglect or obstruct the 5Rs (typically because of copyright restrictions) (Wiley and Hilton, 2018). It can be useful then to

300 consider how "openness' can be understood and assessed. McNally and Christiansen (2019) suggest OER openness can be evaluated based on the eight primary factors, including copyright, accessibility, language, support costs, assessment,

digital distribution, file format, and cultural considerations. They experiment with using these criteria with a three-part scale (closed, mixed, most open) - meaning that resources may be "most open" with regard to some criteria and "mixed" or "closed" in relation to others. Their work suggests that the relative openness of OERs (conceptualized through the 5Rs) can and should be evaluated by educators, as we have done here.

305

In the discussion that follows, we use students' survey responses to  evaluate relative openness/accessibility and other pedagogical factors and  grade the two course modules accordingly through an *Open Enough* rubric (Christiansen and McNally, 2022) (Table 2; treating *Harvestability* as a *Technical* rather than *Pedagogical* factor). Both Geo-UAV and Geo-SfM (and the Jupyter Book/GitHub framework as a whole) rank high on openness, outranking many of those rated by Christiansen and McNally (2022). Key considerations are the modules' learner-centred design and the implementation of collaborative design choices.

310

Although interactivity First, the interactivity of the modules, exposure to pre-written code (snippets), and integrated multimedia use provided a rich and diverse learning experience certainly helped demystify the abstract notions of scientific data acquisition and processing, . Second, we noticed that the availability of co-creation examples and introductions to the unformatted from previous years to learn from, as well as being introduced to the unformatted source code of the teaching resources lowered the

325  barrier for students to become contributors. Students affirmed as much and specifically noted the efficacy of Third, students noted the learning effectiveness of the modules through their step-by-step instructions that were provided in various formats, different voices, and different levels of interactivity. Students also Fourth, students affirmed what we had hypothesised – that for students to become

clarity, and ease of use. Both modules generally rated positively on diversity of content, navigation, and their modern design, though would benefit from being translated into additional languages . Perhaps the most important reflections came on the use and integration of animations and videos in addition to the rich text descriptions, which were stated to greatly benefit the diversity and accessibility of the course content. Where shorter animations of up to a few seconds were preferred to explain single steps, students seemed to prefer pausable videos for content with longer playtimes that covered multistep processes. During

340  plenum discussions, students largely agreed with our hypothesis that videos form a higher participation-barrier for co-creation, especially given the ease with which short animations can be re-recorded and updated, and higher cost of videos in terms of time, IT skills, and storage requirements. Thus, in addition to being low-bandwidth, animated gifs were found to be ideally suited as long as the content was sufficiently decomposed into digestible chunks. Further studies are, however, needed to ascertain these findings and find optimal playtime durations for animated and video content.

(improving upon the modules' current *Closed* language rating in the *Open Enough* rubric, Table 2). Indeed, some of the

345  technologies and software being used were nascent and unfamiliar to students, though this was easily overcome through active

**4.2 Design choices - lessons learned and future directions**

how to design and co-create future OER-P content.

 As instructors, it is encouraging to see that  off-the-shelf software and infrastructure  have reached the point where open-source curricula can be easily created, shared, adapted, and, importantly, used and found  (e.g., Chen and Asta, 2022; Kim et al., 2021; Executable Books Community, 2020). With these tools, instructors and learners alike have access to and can remix different compendium versions and works for their course-specific  (learning) needs. Moreover, using for example the GitHub backend, adaptations can be easily tracked  and reintegrated where applicable with the source, alternatively form the foundation for derived educational content. Indeed,  in doing so, such adaptations and co-creations contribute to the community-driven development of OERs that makes learning more accessible (e.g., Kim et al., 2021).

possible point of concern, with students and practitioners often unaware of existing modules developed elsewhere , as for example dedicated modules on Python and GIS (e.g., Henrikki Tenkanen et al., 2022; QGIS Project, 2024). The latter, in particular, became evident from student responses that requested additional compendiums on GIS and programming,  Herein tutoring plays an important role, at least where it concerns aiding students with how to best find (educational) resources online, and it may be beneficial to establish curated (and searchable) portals for thematic content.

should be done with caution, as reflected on in Course 2 evaluations by students. Extensive Students in particular mentioned the extensive cross-linking between the Geo-UAV and Geo-SfM modules was often mentioned as a point of confusion , and it may thus be better to integrate, rather than link, (Table 1), underlining the importance to sort corresponding materials in the correct pedagogical structure.

Course 2 also illustrated that the chosen JupyterBook /GitHub framework worked well for both in- and outdoor settings. The It is important to note that students benefited from extended introductions to the JupyterBook interface. This became clear to us only after specific feature requests such as the implementation of a search bar were made, even though search and a search bar is automatically included in the JupyterBook menubar. Here it sits next to buttons used for raising issues and generating portable PDF documents, which students frequently used to contribute and when exporting notes to bring along in the field. The latter was best observed during Course 2. The use of 
[revised manuscript text omitted]

---

## Author Response (AR1)

Dear Mathew Stiller-Reeve,

Thank you for the constructive feedback to the manuscript. Many of the comments were previously addressed and implemented in response to the reviewer comments, and we refer to our previous responses in the open discussion for their implementation, e.g., https://doi.org/10.5194/gc-2024-6-AC1 and https://doi.org/10.5194/gc-2024-6-AC2. In addition, please find below our inline responses to the constructive comments raised in the editor decision.

Please also note that the (latexdiff) manuscript marked with differences is compared to the original submission, not to the snippets provided to the reviewers in the respective replies in the discussion.

With best regards and on behalf of the authors,

Peter Betlem

**(Editor decision comments in white, author response in red)**

The reviewers have already highlighted some issues I also noticed in the paper. Here is a list of these issues:

1. Introduction and Framing: The introduction is too long and focuses too much on the broader debate regarding Open Educational Resources (OERs). Since the paper's focus is the use of Jupyter books, please ensure this is clearly the main topic. Additionally, I would appreciate more background information on Jupyter books themselves, as I am not familiar with them.

> We have restructured the main text to clarify and better emphasise the use of Jupyter Books. We have also provided additional background information on Jupyter Books themselves in the introduction.

2. Research Question: As noted by reviewers, the research question was unclear to me as well. I highly recommend clearly stating a research question (with a question mark) in the introduction, using it as an anchor throughout the paper. The readership will benefit if you provide a concrete answer to this question at the beginning of the Discussion section before elaborating on the results.

➢ We have restructured the introduction and expanded the last introduction paragraph to state the goals of the paper, then discussed them in the discussion. We have also shortened the introduction.

3. Figures and Tables: I found Figure 1 challenging to understand due to the arrows pointing in various directions and the boxes labeled "course n," which do not relate to this particular research story. The figure does not aid my understanding of the text. Please either remove it or revise it (and the text) so that it adds value for the reader. Table 1 requires tidying to prevent overlapping text. Table 2 seems tangential, as it introduces "Open enough Rubric" for the first time. Table 2 presents results from an analysis, which should be described in the Methods, presented in the Results, and discussed in the Discussion. It should also clarify how this new information helps answer the research question you pose.

➢ Table 1 was previously updated while addressing R2 and the suggested changes implemented in this latest version.
➢ We have revised Figure 1 and improved its caption as well as provided additional details in the main text.
➢ Table 2 was updated/moved as part of the response to R1. Additional brief sections were added to the methods and results sections to explain and detail the method and findings.

4. Ethical Considerations: You mention that anonymity was maintained, which is good. However, please reference the ethical guidelines you followed or any approvals you received. It is important to ensure the survey was conducted ethically and that data storage and management were appropriate. This needs to be explained in the paper. Please refer to this article (https://gc.copernicus.org/articles/4/493/2021/) for advice, or feel free to ask me for further clarification.

➢ A link to the ethical guidelines was added to the methods section and the relevant paragraph was expanded with additional information about the study design.

5. Interdisciplinary Group: You mention that natural scientists, social scientists, and students need to be involved in the developments you describe. Could you please provide details on how your team was composed of natural and social scientists and how this collaboration worked?

> ➢ We provide a brief overview of how the team was composed and how this collaboration worked below. Some additions were made to the main text, but please let us know to what extent this should be further addressed.
>   - o Natural scientists (PB and NR) implemented the first versions of the modules.
>   - o In dialogue with social scientist (MVK), the modules were updated, and the questionnaire was developed to obtain quantitative and qualitative student feedback. Discussions and pedagogical insights provided by MVK further improved accessibility for students.
>   - o Co-creation was pursued through a shared interest of the natural scientists, familiar with code versioning and co-creation in programming projects, and MVK, who highlighted the benefits of the approach in teaching. Students were encouraged by the instructors to actively contribute to the resources through the back end.
>   - o Logistics support (SMC) provided practical feedback from a logistics point of view, and technical feedback related to operational design, suggesting the Geo-MOD course (Course 2) to bring attention to the usability of co-creation and shared resources across departments.

Above all, please ensure that the real focus is on the Jupyter book you developed and the implementation and assessment efforts that made it beneficial for your students. I look forward to seeing the updated version. Please implement the changes you have suggested to the reviewers and respond to my comments by making changes directly to the text.

> ➢ All changes suggested to the reviewers have been implemented. These have been further revised with regard to the constructive comments and suggestions raised above.

---

## Editor Decision (ED1)

Thank you for providing the updated article. You have made significant progress in addressing many of the points raised in the first round of review. However, there are still some areas that require further work, as several issues from the previous round remain unresolved. I will also address finer details at the sentence level of your article.

1. **Introduction and Framing**: The introduction is now much crisper and more focused.
2. **Goals in the Introduction**: These are also communicated more clearly.
3. **"Open Enough" Rubric**: I see that you've added brief sections in the Methods and Results to explain this element of your study. However, more information is needed. You dedicate considerable time to discussing these results, so we need more context on how they were derived and what they mean. I still have questions about how you define the different classifications. For example, what is the difference between "most open" and "mixed"? What does it mean that Harvestability is a technical rather than pedagogical factor? How did you evaluate that the courses were culturally inclusive? There must be a systematic method behind these results that should be described in greater detail.
4. **Interdisciplinary Group**: You provided details about how the interdisciplinary group was composed in your response, but this information is missing from the article. Please include this in the article. Currently, you mention that the project "can only be accomplished through an interdisciplinary collaboration between scientists, social scientists, and students" for the first time in the discussion section and then again in the conclusion. All elements in these sections should have a foundation earlier in the paper. If this aspect is important to your argument, it needs to be detailed earlier; if not, it can be left out.
5. **Methods in the Abstract**: Please make the Abstract more concise to allow space to briefly explain the methods you used to evaluate your courses. Also, the final sentence of an Abstract is typically the take-home message. It seems unlikely that your main takeaway for this paper is about GIFs. The second-to-last sentence functions much better as a concluding message.

Otherwise, I highly recommend going through every sentence to see if you can make them more concise and to-the-point. There are quite a number of heavy sentences that this reader needed to re-read several times to understand the point or still did not understand. Here are a few examples. If I do not include an explanation then it simply means that it was difficult to read and needs revision for clarity and consiseness:

*Behind the scenes, was used to facilitate content versioning, co-creation and open publishing of the resources.*

*Herein OERs and OER-enabled pedagogy (OER-P) play an important role and have seen an update in recent years as an alternative to conventional scholarly and educational publishing.*

*Recently, the Jupyter Book environment has emerged as an extention that extends the computational Notebook environment with narrative and multimedia content.*

*In this contribution, we test whether Jupyter Books can indeed act as a diverse, equitable, and inclusive learning environment, embracing the three pillars of "open" social justice: redistributive, recognitive, and representational.*

- This was a particularly challenging sentence. Do this pillars correspond with the "first", "second" and "third" in the proceeding sentences? If so, we need to know what "redistributive, recognitive, and representational" mean in this regard. Also, you say you "test" these three pillars, but you never mention them again. Please consider deleting this if it does not impact the story further on.

*It then demonstrates the openness and accessibility of the framework (including the use of animations), assesses user and student learning experiences, and appraises the framework's co-creation possibilities.*

- A paper can document something, but it cannot demonstrate or assess something. You, the authors did these things.

*This study was conducted over 4 years as part of two geology courses at the University Centre in Svalbard, a small public university centre in northern Norway.*

- Svalbard is not in northern Norway. It is an archipelago in the Arctic Ocean.

*Subsequent years saw in-person teaching with minor revisions based on colloquial and questionnaire feedback.*

- What is a "colloquial" questionnaire? Colloquial means informal, everyday language. Please ensure that this is described more accurately, both here and other places you use the term.

*Fieldwork tested the portability of Geo-UAV, implementing either the online tutorials or exported PDFs while teaching in the field.*

*Classroom teaching further implemented the colloquial sharing of results and experiences during daily recaps in which students presented both their results and stumbles, with feedback and possible solutions mostly provided by other working groups.*

- Classroom teaching? But in the methods you say "Both courses were taught asynchronously during a one-week interval by the same instructors, with all materials provided online." Are the courses asynchronous or not? Please ensure consistency here. Also, you use the word colloquial again. I'm sure you don' t mean informal and everyday language.

*However, we evaluated Harvestability as a Technical rather than Pedagogical factor, and based the ranking on a combination of colloquial and questionnaire student feedback, as well as on our own observations as educators and instructors.*

- What does Harvestability mean? What is colloquial feedback? Please see comment about this method earlier.

*Course 1 (n=30) and at the end of Course 2 (n=10)*

- But you mention earlier than "n=62 over four years". Please ensure consistency with these numbers.

*Unsurprisingly, a subset of students in prior years reported agreement that they were "a bit confused … when it came to using " as they were not fully introduced to the platform's possibilities at the onset of the courses. The differing levels of introduction, however, did not change student-reported inclusiveness in content creations, or their overall learning experience.*

*Fourth, students affirmed what we had hypothesised– that for students to become contributors, they first need to be comfortable using the tools and be given ample opportunity and freedom to revise content, with the side note that it is reviewed and fact-checked by other students and course instructors prior to implementation.*
- You hypotheiszed? I do not recall you doing this.

*Using Jupyter Book/, changes in course content can be easily tracked and reintegrated where applicable with the source or form the starting point for derived educational content, contributing to the community-driven development of OERs that makes learning more accessible.*

*… seems to have lowered the participation barrier which may also benefit non-traditional learners and students from underrepresented groups who may have less initial experience with either of the topics covered by the modules.*
- This idea about non-traditional learns or students from underrepresented groups is conjecture and needs to be deleted, unless you have evidence to back up these claims.

---

## Author Response (AR2)

Dear Mathew Stiller-Reeve,

Thank you for providing us with further constructive feedback. We have gone through the entire manuscript and significantly improved its general readability. In addition, please find below our inline responses to the comments raised as part of the editor decision. We hope the proposed revisions address the raised and any other remaining concerns.

Please also note that the (latexdiff) manuscript marked with differences is compared to the previous revision, not to the original manuscript nor the snippets provided to the reviewers in the respective replies in the open discussion.

With best regards and on behalf of the authors,
Peter Betlem

**(Editor decision comments in white, author response in red)**

Thank you for providing the updated article. You have made significant progress in addressing many of the points raised in the first round of review. However, there are still some areas that require further work, as several issues from the previous round remain unresolved. I will also address finer details at the sentence level of your article.

1. **Introduction and Framing**: The introduction is now much crisper and more focused.
   Further modifications have been made to the introduction as part of the general readability improvements.

2. **Goals in the Introduction**: These are also communicated more clearly.

3. **"Open Enough" Rubric**: I see that you've added brief sections in the Methods and Results to explain this element of your study. However, more information is needed. You dedicate considerable time to discussing these results, so we need more context on how they were derived and what they mean. I still have questions about how you define the different classifications. For example, what is the difference between "most open" and "mixed"? What does it mean that Harvestability is a technical rather than pedagogical factor? How did you evaluate that the courses were culturally inclusive? There must be a systematic method behind these results that should be described in greater detail.
   Following further considerations we have decided to remove the Open Enough rubric. While we initially deemed it a useful visualization method to summarise the modules' openness, we realise now its inclusion takes away the focus from the key findings of the study.

4. **Interdisciplinary Group**: You provided details about how the interdisciplinary group was composed in your response, but this information is missing from the article. Please include this in the article. Currently, you mention that the project "can only be accomplished through an interdisciplinary collaboration between scientists, social scientists, and students" for the first time in the discussion section and then again in the conclusion. All elements in these sections should have a foundation earlier in the

paper. If this aspect is important to your argument, it needs to be detailed earlier; if not, it can be left out.

Added a paragraph on the interdisciplinary flow in the "Module and course design" section. We also softened the phrase to read "Such an effort certainly benefits from…"

5. **Methods in the Abstract**: Please make the Abstract more concise to allow space to briefly explain the methods you used to evaluate your courses. Also, the final sentence of an Abstract is typically the take-home message. It seems unlikely that your main takeaway for this paper is about GIFs. The second-to-last sentence functions much better as a concluding message.

The abstract has been shortened while brief statements on the methods have been added.

Otherwise, I highly recommend going through every sentence to see if you can make them more concise and to-the-point. There are quite a number of heavy sentences that this reader needed to re-read several times to understand the point or still did not understand. Here are a few examples. If I do not include an explanation then it simply means that it was diHicult to read and needs revision for clarity and consiseness:

Our sincerest apologies. It seems the word GitHub was removed during final typesetting/generation of the PDF, causing many of the incomplete and unclear sentences referred to. Text occurrences of GitHub have been restored.

*Behind the scenes, was used to facilitate content versioning, co-creation and open publishing of the resources.*
See previous comment about GitHub accidentally being removed in typesetting.

*Herein OERs and OER-enabled pedagogy (OER-P) play an important role and have seen an update in recent years as an alternative to conventional scholarly and educational publishing.*

*Paragraph has been partly rewritten, and sentences shortened.*

*Recently, the Jupyter Book environment has emerged as an extention that extends the computational Notebook environment with narrative and multimedia content.*

*Paragraph has been partly rewritten, and sentences shortened.*

*In this contribution, we test whether Jupyter Books can indeed act as a diverse, equitable, and inclusive learning environment, embracing the three pillars of "open" social justice: redistributive, recognitive, and representational.*
- This was a particularly challenging sentence. Do this pillars correspond with the "first", "second" and "third" in the proceeding sentences? If so, we need to know

what "redistributive, recognitive, and representational" mean in this regard. Also, you say you "test" these three pillars, but you never mention them again. Please consider deleting this if it does not impact the story further on.

The pillars have been removed.

*It then demonstrates the openness and accessibility of the framework (including the use of animations), assesses user and student learning experiences, and appraises the framework's co-creation possibilities.*
- A paper can document something, but it cannot demonstrate or assess something. You, the authors did these things.

Agreed. We have modified and clarified the paragraph further.

*This study was conducted over 4 years as part of two geology courses at the University Centre in Svalbard, a small public university centre in northern Norway.*
- Svalbard is not in northern Norway. It is an archipelago in the Arctic Ocean.

This has been modified and future references to the centre have been replaced by its acronym (UNIS).

*Subsequent years saw in-person teaching with minor revisions based on colloquial and questionnaire feedback.*
- What is a "colloquial" questionnaire? Colloquial means informal, everyday language. Please ensure that this is described more accurately, both here and other places you use the term.

Both here and elsewhere "colloquial" has been replaced with in-class feedback, classroom discussions, or similar.

*Fieldwork tested the portability of Geo-UAV, implementing either the online tutorials or exported PDFs while teaching in the field.*

This sentence has been removed.

*Classroom teaching further implemented the colloquial sharing of results and experiences during daily recaps in which students presented both their results and stumbles, with feedback and possible solutions mostly provided by other working groups.*
- Classroom teaching? But in the methods you say "Both courses were taught asynchronously during a one-week interval by the same instructors, with all materials provided online." Are the courses asynchronous or not? Please ensure consistency here. Also, you use the word colloquial again. I'm sure you don' t mean informal and everyday language.

This has been reworded for clarity: "Both courses were taught and applied asynchronously throughout the semester, with physical tutoring hours available over a one-week period. All materials were provided online, and follow-up discussions taking place both digitally and in person."

*However, we evaluated Harvestability as a Technical rather than Pedagogical factor, and based the ranking on a combination of colloquial and questionnaire student feedback, as well as on our own observations as educators and instructors.*

-What does Harvestability mean? What is colloquial feedback? Please see comment about this method earlier.

This section has been removed. See previous comments.

*Course 1 (n=30) and at the end of Course 2 (n=10)*

- But you mention earlier than "n=62 over four years". Please ensure consistency with these numbers.

This has been reworded to clarify that n=30 applies to 2023 and 2024.

*Unsurprisingly, a subset of students in prior years reported agreement that they were "a bit confused ... when it came to using " as they were not fully introduced to the platform's possibilities at the onset of the courses. The di^ering levels of introduction, however, did not change student-reported inclusiveness in content creations, or their overall learning experience.*

See previous comment about GitHub accidentally being removed in typesetting.

*Fourth, students a^irmed what we had hypothesised– that for students to become contributors, they first need to be comfortable using the tools and be given ample opportunity and freedom to revise content, with the side note that it is reviewed and fact-checked by other students and course instructors prior to implementation. -* You hypotheiszed? I do not recall you doing this.

This section has been rephrased.

*Using Jupyter Book/, changes in course content can be easily tracked and reintegrated where applicable with the source or form the starting point for derived educational content, contributing to the community-driven development of OERs that makes learning more accessible.*

See previous comment about GitHub accidentally being removed in typesetting.

*… seems to have lowered the participation barrier which may also benefit nontraditional learners and students from underrepresented groups who may have less initial experience with either of the topics covered by the modules*.

- This idea about non-traditional learns or students from underrepresented groups is conjecture and needs to be deleted, unless you have evidence to back up these claims.

This section has been rephrased.

---

## Author Response (AR3)

Dear Editors Solmaz Mohadjer and Mathew Stiller-Reeve,

Thank you for the feedback throughout the review process.

We attach the corrected manuscript which implement the requested technical corrections:

- In-line -> online (line #3, Abstract)
- Listed 5Rs (line #21, Introduction)
- Spelled out UAV and SfM (line #5-6, Abstract; line #46, Introduction)

Thank you once again for helping us improve our work.

With best regards, and on behalf of the authors,

Peter Betlem